# Gelatinase regulates the egress of intracellular replicating populations during *Enterococcus faecalis* infection

Frederick Reinhart Tanoto[1,2], Jia Hui Liew[3], Claudia J. Stocks[1,4], Deepti Rawat[5], Kelvin Kian Long Chong[1], Kevin Pethe[1,6,7,8], Haris Antypas[1], Kimberly A. Kline[1,3,5,7]*

1 Singapore Centre for Environmental Life Sciences Engineering, Nanyang Technological University, Singapore, Singapore, 2 Interdisciplinary Graduate Programme, Nanyang Technological University, Singapore, Singapore, 3 School of Biological Sciences, Nanyang Technological University, Singapore, Singapore, 4 Institute for Molecular Bioscience, The University of Queensland, Brisbane, Australia, 5 Department of Microbiology and Molecular Medicine, Faculty of Medicine, University of Geneva, Geneva, Switzerland, 6 Lee Kong Chian School of Medicine, Nanyang Technological University, Singapore, Singapore, 7 Singapore-MIT Alliance for Research and Technology Centre, Singapore, Singapore, 8 National Center for Infectious Diseases (NCID), Singapore, Singapore

* Kimberly.Kline@unige.ch

## Abstract

*Enterococcus faecalis* is a common opportunistic pathogen, frequently isolated from chronic wounds, yet the mechanisms underlying its virulence and persistence in this niche remain incompletely understood. We previously showed that a subpopulation of *E. faecalis* can survive intracellularly for several days during murine wound infection and can replicate within macrophages, revealing an unexpected intracellular phase for this traditionally extracellular bacterium. Here, we identify the secreted metallo-protease gelatinase (GelE) and its regulator, the Fsr quorum sensing system, as key modulators of *E. faecalis* intracellular survival and replication. Mechanistically, Fsr quorum sensing is induced during intracellular replication, promoting GelE-dependent host cell lysis and bacterial egress. In the absence of active GelE, *E. faecalis* accumulates as large intracellular clusters, a phenotype observed consistently across GelE-deficient wound isolates. In a mouse wound model, GelE-deficient *E. faecalis* similarly exhibited higher intracellular numbers within wound infection-associated host cells. Together, our study uncovers GelE as a central effector that orchestrates the transition between intracellular and extracellular lifestyles of *E. faecalis*, providing a possible explanation for its persistence in chronic wound infection.

## Author summary

Pathogenic bacteria are traditionally classified as either "intracellular" or "extracellular", but growing evidence suggests that many extracellular bacteria also adopt transient intracellular lifestyles that promote persistent and recurrent

**Data availability statement:** Whole genome sequencing data of E. faecalis wound isolates presented in this study are publicly available from NCBI Sequence Read Archive (SRA) with the identifier PRJNA1369111. Original uncropped, labelled Western blot is shared as S1 Data. All other relevant data are within the manuscript, with raw values provided as S2 Data.

**Funding:** Funding for this work was provided by the National Research Foundation and Ministry of Education Singapore under its Research Centre of Excellence Program to KAK, by the Ministry of Education Singapore under its Tier 2 programs to KAK (MOE2018-T2-001-127 and MOE2019-T2-002-089), by Swiss National Science Foundation (SNSF) to KAK (310030_212262), and by SCELSE Seed Funding (SF-05) to HA. FRT is supported by the Interdisciplinary Graduate Programme, Nanyang Technological University and the Nanyang President's Graduate Scholarship, Nanyang Technological University. The funders had no role in study design, data collection and analysis, decision to publish, or preparation of the manuscript.

**Competing interests:** The authors have declared that no competing interests exist.

infection. *Enterococcus faecalis*, a leading cause of chronic wound infection, exemplifies this duality. We discovered that *E. faecalis* strains lacking the secreted protease gelatinase accumulate to high numbers inside host cells such as macrophages. Our data indicate that gelatinase facilitates bacterial escape following intracellular replication, regulating the transition between an intracellular and extracellular lifestyle. During infection, gelatinase-deficient bacteria remain hidden within various wound cell types, potentially evading immune clearance and antibiotic treatment. This work reveals a previously unrecognized role for gelatinase in controlling *E. faecalis* intracellular dynamics, highlighting a mechanism that may underline chronic and persistent infection.

## Introduction

*Enterococcus faecalis* is an opportunistic pathogen associated with chronic wounds, infective endocarditis, catheter-associated urinary tract infections (CAUTI), and bacteremia [1,2]. Although traditionally viewed as an extracellular organism [3], accumulating evidence indicates that *E. faecalis* can persist intracellularly in diverse host cells, including endothelial, epithelial, and immune cells such as macrophages and neutrophils [4–10]. *In vivo,* intracellular *E. faecalis* has been detected in the gastrointestinal tract, liver, and infected wound tissue [6,7,11,12], where they may act as reservoirs enabling long-term colonization, dissemination, or reactivation to cause recurrent infection [11,13,14]. Evidence of intracellular replication in both immune and non-immune cells [6,10,12] suggests that *E. faecalis* intracellular persistence is an active, regulated process. Identifying the bacterial determinants that enable survival and replication within host cells is therefore critical to understanding *E. faecalis* pathogenesis.

Multiple determinants contributing to *E. faecalis* intracellular survival have been described. The multiple-peptide resistance factor MprF2 protects against antimicrobial peptides [15], while methionine sulfur reductase MsrB and thiol peroxidase Tpx detoxify reactive oxygen species [16,17]. Phosphotransferase systems PTS8 (encoded by OG1RF_12399 to OG1RF_12402) and *mptABCD* (PTS1) appear to modulate intracellular metabolism and macrophage reprogramming, and several transcriptional regulators influence intracellular persistence, although their effectors are unknown [18–23]. *E. faecalis* can also delay phagolysosomal fusion, damage phagosomes, and evade autophagic clearance [6–8], reflecting its capacity to adapt within host cell. Additional virulence factors that mediate colonization or immune modulation may also support intracellular persistence. The adhesin Ace promotes host cell attachment and survival in macrophages [24], lactate dehydrogenases (Ldh) suppress NF-kB-dependent macrophage activation [25] and the type VII secretion system (T7SS) may likewise secrete effectors that enhance intracellular survival [26], similar to that of *Staphylococcus aureus* and other secretion systems from intracellular Gram-negative bacteria [27–30]. Two-component systems (TCS) further enable *E. faecalis* to sense and adapt to dynamic environments and have been implicated

in intracellular survival [31,32]. Among these is the well-characterized quorum sensing (QS) FsrABDC system [33,34], comprising a response regulator FsrA, a sensor kinase FsrC, and a membrane protein FsrB which processes FsrD, a precursor to the autoinducer peptide gelatinase biosynthesis-activating pheromone (GBAP). Altogether, FsrABDC regulates the secreted metalloprotease gelatinase (GelE), serine protease (SprE), and enterococcin V (EntV), all previously linked to virulence but not to intracellular behavior [35–38]. Given the confined environment of the host cell interior, replicating bacteria may activate QS system inside the host cell, although this possibility or its consequence has not been explored. Together, these studies highlight the multifactorial nature of *E. faecalis* intracellular adaptation and the need to define the regulatory pathways coordinating these processes.

To address these gaps, we examined candidate *E. faecalis* genes for roles in intracellular persistence using an *in vitro* macrophage infection model. Our screen identified GelE and its regulatory Fsr QS system in regulating intracellular *E. faecalis* across diverse clinical wound isolates. We found that *fsrABDC* and its associated regulon are upregulated during intracellular replication starting at 6 hours post-infection (hpi), promoting bacterial egress and regulating the intracellular population density of replicating *E. faecalis*. Consistent with these findings, ΔgelE strains persisted intracellularly in immune and non-immune cells in a mouse wound infection model at 5 days post-infection (dpi). Together, these results identify QS-regulated GelE as a central factor controlling the intracellular-extracellular balance of *E. faecalis* and demonstrate the Fsr QS system as a key regulator of this transition during infection.

## Results

### Absence of *fsrA* and *gelE* enhances intracellular survival of *E. faecalis* in RAW264.7 macrophages

To investigate how *E. faecalis* regulates intracellular survival, we screened a panel of *E. faecalis* mutants of genes associated with intracellular persistence or overall virulence for their ability to survive intracellularly within RAW264.7 macrophages. These included transposon insertion mutants from the OG1RF EfaMarTn library [39] and in-frame deletion mutants (S1 Table). Macrophages were infected at an MOI of 10 for 1 h, followed by antibiotic treatment to rapidly eliminate extracellular bacteria in the absence of significant host cell cytotoxicity (S1A–S1B Fig), and viable intracellular bacteria were quantified at 2, 6, and 20 hpi (S1C–S1D Fig). We did not observe significant differences in intracellular colony forming units (CFU) for any mutant at 2 hpi (S1E Fig). However, at 6 hpi when intracellular CFU peaked (S1C–S1D Fig), ΔmprF2 and OG1RF 12401::Tn (encoding enzyme IIC in PTS8) showed significantly fewer intracellular CFU compared to WT (Fig 1A). At 20 hpi, both ΔmprF2 and OG1RF_12401::Tn continued to show significantly fewer intracellular CFU, as did *msrB*::Tn and ΔmptD (Fig 1B), corroborating previous studies implicating each of these genes in survival within macrophages [15,16,18,20]. Surprisingly, we also found that virulence mutants *fsrA*::Tn and *gelE*::Tn accumulated to significantly higher intracellular CFU compared to WT (Fig 1B). Because FsrA regulates the transcription of *gelE*, *sprE*, and *entV* [33,34] and GelE proteolytically activates downstream targets including Ace, AtlA, and EntV [40–42], we examined whether loss of these downstream effectors might account for the increased intracellular survival. In contrast to *gelE*::Tn, transposon insertion mutants of *ace, atlA*, and *entV* displayed similar intracellular CFU as WT at 20 hpi. These data suggest that FsrA and GelE, rather than other downstream effectors of the Fsr regulon, mediate the modulation of intracellular survival.

To validate these findings, we generated in-frame deletion mutants for individual transcriptional units (ΔfsrA, ΔfsrBDC, ΔgelE and a combined transcriptional unit ΔfsrABDC; Fig 1C) and quantified their intracellular CFU. We additionally tested a transposon insertion mutant in *sprE*, a serine protease encoded downstream of *gelE* that is also regulated by FsrA, for its contribution to intracellular survival. All mutants had similar intracellular CFU at 2 hpi (S1F Fig). However, ΔgelE and ΔfsrABDC displayed increased intracellular CFU than WT at both 6 and 20 hpi, and ΔfsrA displayed increased intracellular CFU at 20 hpi (Fig 1D–1E). These phenotypes correlated with loss of GelE secretion as confirmed by agar-based activity assays and Western blot of culture supernatants collected at different growth timepoints (Figs 1F–H and S2A). WT initially secreted a 34.5 kDa isoform of GelE with lower activity at 4 h that was progressively cleaved to a 33 kDa fully-active form

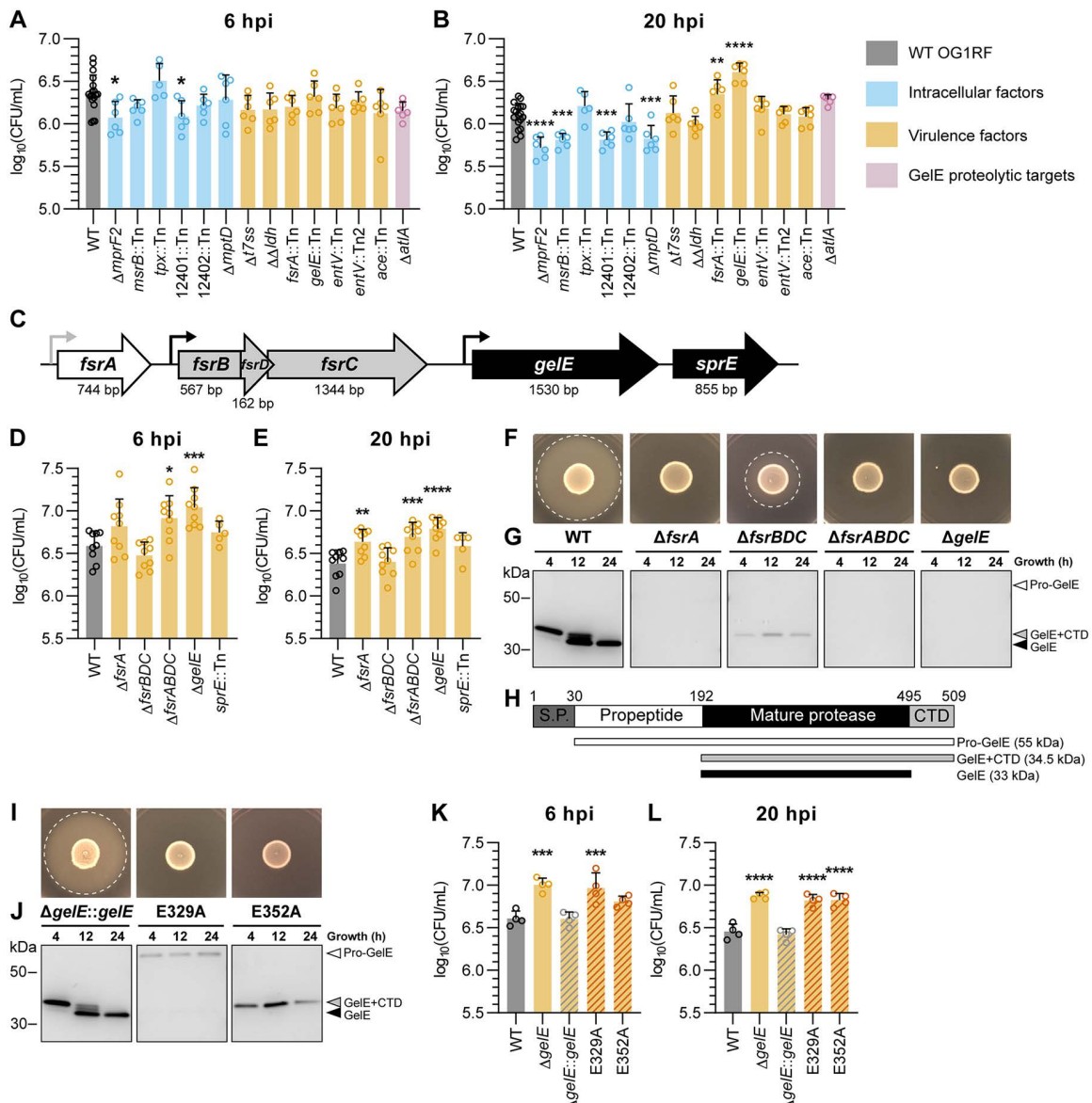

**Fig 1. Absence of *fsrA* and *gelE* enhances intracellular survival of *E. faecalis* in RAW264.7 macrophages at 20 hpi. (A-B)** Intracellular CFU in RAW264.7 macrophages infected with OG1RF WT and derived mutants of genes implicated in intracellular persistence (blue), virulence (orange), or GelE proteolytic targets (pink). (n = 18 for WT and n = 5-6 for mutants) **(C)** Schematic diagram of the genomic locus comprising *fsrABDC/gelE/sprE*, represented in different shaded arrows for each transcriptional unit. Promoters are represented by bent arrows (grey: constitutive promoter of *fsrA*, black: *fsrA*-regulated promoter). The respective gene size in base pairs (bp) is indicated below each gene. **(D-E)** Intracellular CFU in RAW264.7 macrophages infected with genetic deletion mutants of *fsrABDC* and *gelE* (n = 9), as well as a transposon insertion mutant of *sprE* (n = 5). **(F, I)** Gelatinase activity phenotype. **(F, I)** Gelatinase activity of *E. faecalis* strains on Todd-Hewitt agar + 3% gelatin at 24 h. Halo formation (white dashed line) indicates gelatinase activity. Representative images of n = 3 are shown. **(G, J)** Detection of GelE secretion in *E. faecalis* culture supernatants at 4, 12 and 24 h by Western blotting. CTD = C-terminal domain. Representative images of n = 2 are shown. **(H)** Schematic diagram of GelE domains: a putative signal peptide (S.P., 1-30 a.a.), a propeptide (30-192 a.a.), the mature GelE protease (192-495 a.a.) and a C-terminal domain (CTD, 495-509 a.a.). **(K-L)** Intracellular CFU in RAW264.7 macrophages infected with WT, Δ*gelE* and Δ*gelE*-complemented OG1RF strains. (n = 4); All bar graphs shown represent mean ± SD of biological replicates. Statistical significance of each strain against WT was assessed using one-way ANOVA with Dunnett's multiple comparisons test. Only comparisons with p < 0.05 are annotated. * = p < 0.05, ** = p < 0.01, *** = p < 0.001, **** = p < 0.0001.

by 24 h, consistent with previous observations [43]. By contrast, Δ*fsrBDC* retained weak gelatinase activity and had intracellular CFU comparable to WT, consistent with low-level *gelE* expression, likely driven by constitutive *fsrA* transcription from its own promoter. *sprE*::Tn, which was gelatinase-positive, showed comparable intracellular CFU to WT at all time-points, further supporting a specific role of GelE in modulating intracellular survival (Figs 1D–1E and S1G).

To determine whether augmented intracellular CFU depends on GelE proteolytic activity, we chromosomally complemented Δ*gelE* with either the WT allele (Δ*gelE*::*gelE*) or two catalytically inactive variants carrying E329 or E352 substitutions (S3 Fig). The WT complement restored active GelE secretion, whereas the E329A and E352A mutants produced enzymatically inactive isoforms with impaired secretion particularly for E329A, suggesting differential roles for these residues in GelE proteolytic maturation and secretion (Figs 1H–1J and S2B). Accordingly, Δ*gelE*::*gelE* restored intracellular CFU to WT levels, whereas the E329A and E352A mutants lacking gelatinase activity retained elevated intracellular CFU (Figs 1K–1L and S1H). These findings demonstrate that the ability of GelE to regulate intracellular CFU depends on its proteolytic activity.

### Increased intramacrophage survival in the absence of *fsrA* and *gelE* is conserved across wound isolates of *E. faecalis*

The *fsr* QS locus varies among *E. faecalis* clinical isolates from the gut, infective endocarditis, bloodstream, and urinary tract infections [44–47]. To determine whether elevated intracellular CFU observed for OG1RF Δ*fsrA*, Δ*fsrABDC*, and Δ*gelE* mutants extends to clinical strains, we examined 49 *E. faecalis* wound isolates [48,49]. Whole genome sequence analysis of *gelE*, *fsrA* and *fsrBDC* revealed three genotypes: 22/49 strains (44.9%) had the full *fsrABDC* and *gelE* locus (*fsrABDC*⁺*gelE*⁺), 10/49 (20.4%) lacked both *fsrABDC* and *gelE* (*fsrABDC*⁻*gelE*⁻), and 17/49 (34.7%) had a complete *gelE* gene sequence but lacked *fsrA* and had a truncated *fsrBDC* (*fsrABDC*⁻*gelE*⁺). Phenotypically, *fsrABDC*⁻*gelE*⁻ and *fsrABDC*⁻*gelE*⁺ strains lacked gelatinase activity on gelatin agar (GelE⁻), whereas *fsrABDC*⁺*gelE*⁺ strains retained it to varying degrees (GelE⁺) (Figs 2A and S4B). All *fsrABDC*⁻*gelE*⁺ strains shared a conserved 23.9 kb deletion spanning the 5' end of *fsrC* (junction at position 261 in *fsrC*, V583 coordinates: 1,765,811) to the 3' end of *yqeK* (EF1841, V583 coordinates: 1,789,714) (S4A–S4B Fig), consistent with previous reports [50,51]. Genomic analysis of *fsrABDC*⁻*gelE*⁻ strains showed a different 27.8 kb conserved deletion spanning the 5' end of EF1814 (V583 coordinates: 1,759,390) to the 3' end of EF1838 (V583 coordinates: 1,787,212), while *fsrABDC*⁺*gelE*⁺ strains displayed diversity in the genomic region upstream of *fsrA*, including a previously reported 14.8 kb insertion in 11 out of 22 *fsrABDC*⁺*gelE*⁺ wound isolates as well as in OG1RF [51].

We next infected RAW264.7 macrophages with four gentamicin-susceptible strains from each genotype group (highlighted in red in Figs 2A, S4B, and S5A–S5B), as well as JH2–2, a *fsrABDC*⁻*gelE*⁺ laboratory strain [52]. Except for one *fsrABDC*⁻*gelE*⁻ isolate (20_EF), all clinical strains exhibited comparable intracellular CFU at 2 hpi (Fig 2B). At 6 and 20 hpi, all *fsrABDC*⁻*gelE*⁻ strains, as well as the Δ*fsrABDC* and Δ*gelE* controls, showed significantly higher intracellular CFU compared to WT OG1RF (Fig 2C–2D). By 20 hpi, three of the *fsrABDC*⁻*gelE*⁺ strains (04_EF, 12_EF, JH2–2) also exhibited increased intracellular CFU relative to WT, whereas *fsrABDC*⁺*gelE*⁺ strains maintained WT levels at all time points (Fig 2B–2D). GelE secretion was nearly undetectable in culture supernatants of most of the *fsrABDC*⁻*gelE*⁺ clinical strains tested, suggesting that the variability in their intracellular CFU at 20 hpi is likely due to other genomic factors and not a leaky Fsr-independent GelE expression (S6 Fig). Therefore, we conclude that enhanced intramacrophage CFU in the absence of FsrABDC/GelE is conserved among wound isolates and links naturally occurring *fsrABDC*/*gelE* variants with increased intracellular persistence.

### Lack of GelE promotes the accumulation of replicating intracellular bacterial reservoirs

The elevated intracellular CFU observed in GelE⁻ strains at 20 hpi was not due to differences in initial uptake or infection efficiency (S1E–S1F, S1H, and S7 Figs). To determine if this phenotype reflected increased intracellular replication, *E. faecalis* was pre-stained with the proliferation dye eFluor 670, which is diluted with each division (S8A Fig). Infected

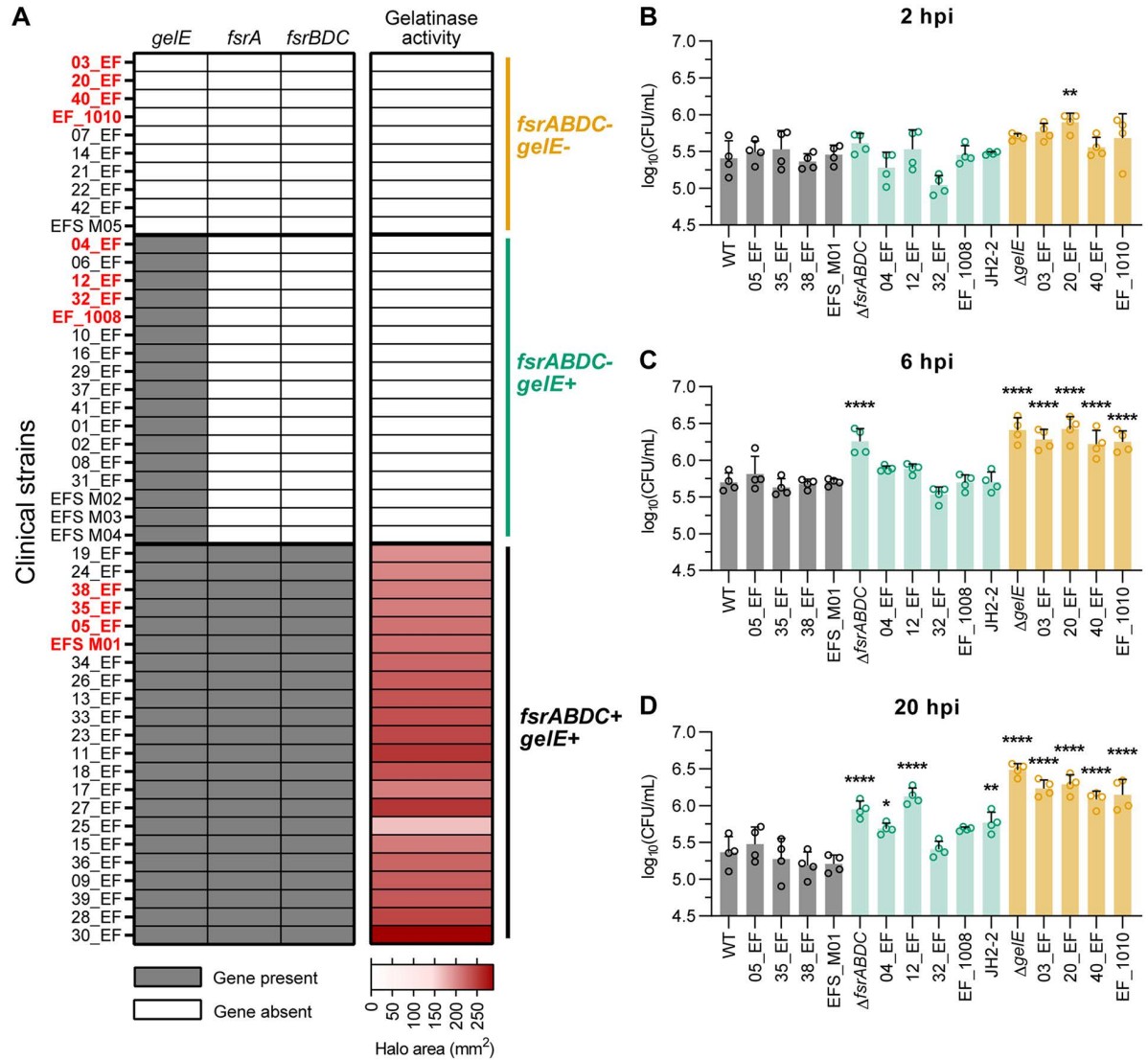

**Fig 2. Increased intramacrophage survival in the absence of *fsrA* and *gelE* is conserved across wound isolates of *E. faecalis*. (A)** Characterization of 49 *E. faecalis* wound isolates for *fsrABDC*/*gelE* genotypes and gelatinase activity phenotypes. **(B-D)** Intracellular CFU in RAW264.7 macrophages infected with four selected gentamicin-susceptible *fsrABDC*+*gelE*+ (grey bars), *fsrABDC*-*gelE*+ (green bars) and *fsrABDC*-*gelE*- (orange bars) *E. faecalis* wound isolates and laboratory strains (JH2-2, OG1RF WT, Δ*fsrABDC* and Δ*gelE*) at **(B)** 2 hpi, **(C)** 6 hpi and **(D)** 20 hpi. Bars represent mean±SD from n=4. Statistical significance of each strain against WT was assessed using one-way ANOVA with Dunnett's multiple comparisons test. Only comparisons with p<0.05 are annotated. * = p<0.05, ** = p<0.01, *** = p<0.001, **** = p<0.0001.

macrophages were lysed at 2, 6, and 20 hpi and intracellular bacteria were analyzed by flow cytometry following immunolabeling of lysates with the anti-Group D antigen (AgD+, labelling all *E. faecalis* cells) (S8B Fig). At 2 hpi, approximately 80–90% of intracellular *E. faecalis* were eFluor+ (Fig 3A–3B, red histograms), indicating limited replication within macrophages regardless of their GelE phenotype. By 6 and 20 hpi, an eFluor- peak of lower fluorescence intensity emerged, indicating intracellular replication (Fig 3A–3B, blue and orange histograms). This eFluor- peak was absent in PFA-fixed non-replicative controls at all timepoints, validating the specificity of this assay in identifying replicating bacteria (Fig 3A). Strikingly, GelE- strains (Δ*fsrA*, Δ*fsrABDC*, Δ*gelE*) displayed significantly higher proportions of replicating eFluor- bacteria

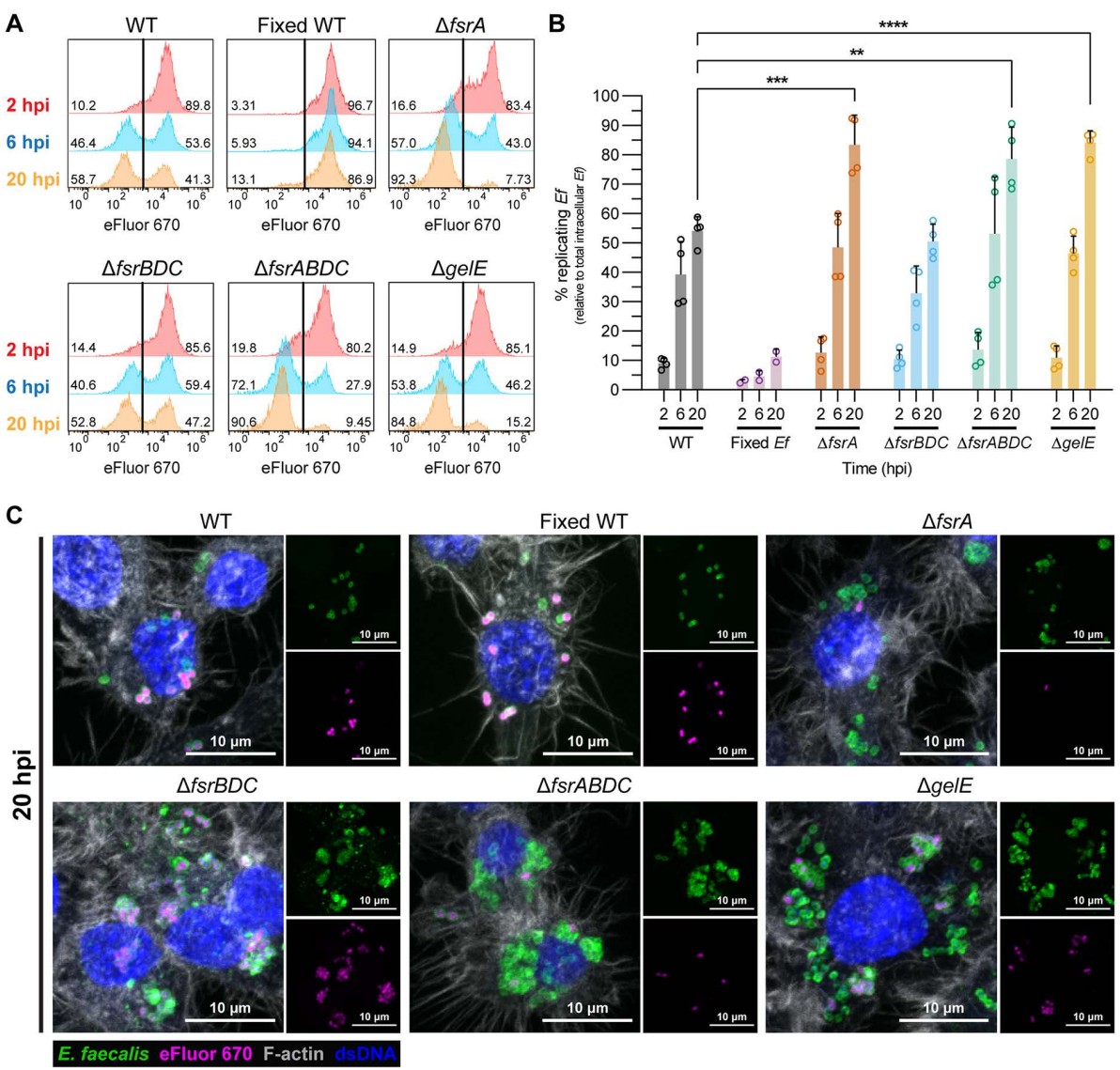

**Fig 3. Lack of GelE promotes the accumulation of replicating intracellular bacterial reservoirs. (A)** Quantification of intracellular replicating (eFluor⁻) and non-replicating (eFluor⁺) *E. faecalis* strains from macrophage lysates at 2, 6, and 20 hpi using flow cytometry. WT OG1RF fixed with 4% PFA prior to infection (Fixed WT) was included as non-proliferating controls. Representative histograms from n = 2-4 are shown. **(B)** Proportion (%) of intracellular replicating *E. faecalis* (eFluor⁻) population from WT and mutant *E. faecalis*-infected macrophage lysates. Bars represent mean ± SD of n = 4, except fixed WT (n = 2). Statistical significance of each strain against WT was assessed using one-way ANOVA with Dunnett's multiple comparisons test. Only comparisons with $p < 0.05$ are annotated. ** $= p < 0.01$, *** $= p < 0.001$, **** $= p < 0.0001$. **(C)** Representative confocal microscopy images of RAW264.7 macrophages infected by *E. faecalis* strains pre-stained with eFluor 670 (magenta) at 20 hpi from n = 3. Samples were fixed and post-stained for *Enterococcus*-specific Group D antigen (green), double-stranded DNA (dsDNA; blue) and F-actin (white).

(~80–90%) compared to GelE⁺ strains (WT and Δ*fsrBDC*), which reached only ~40–60% at 20 hpi. (Fig 3A-3B). Confocal microscopy supported these findings: At 20 hpi, we observed macrophages infected with GelE⁻ strains (Δ*fsrA*, Δ*fsrABDC*, Δ*gelE*) containing large intracellular clusters of eFluor⁻ bacteria, consistent with active replication, whereas infection with GelE⁺ strains (WT, Δ*fsrBDC*) primarily contained eFluor⁺ non-replicating bacteria (Fig 3C). The higher proportion of replicating intracellular GelE⁻ strains was not attributable to increased growth rate, as all *fsrABDC*/*gelE* deletion mutants

exhibited growth kinetics comparable to WT OG1RF when cultured in macrophage culture media (S9 Fig). Collectively, these results demonstrate that the elevated intracellular CFU observed in GelE⁻ *E. faecalis* results from a higher proportion of replicating bacterial reservoirs within macrophages rather than altered uptake or cytotoxicity.

### Intracellularly replicating WT OG1RF induce Fsr quorum sensing at 6 hpi

Since FsrA and GelE negatively regulate intracellular persistence and replication, we next examined whether these genes are differentially expressed in intracellularly replicating GelE⁺ WT OG1RF. To distinguish macrophages containing replicating versus non-replicating intracellular *E. faecalis* at 6 and 20 hpi, we adapted a two-color fluorescence dilution assay combined with fluorescence-assisted cell sorting (FACS) [53,54]. Constitutive Dasher GFP-expressing WT OG1RF were pre-stained with the proliferation dye eFluor 670 to identify macrophages containing replicating bacteria. As intracellular bacteria replicate, the macrophage GFP intensity increases while the eFluor signal remains constant, enabling separation of macrophages containing replicating *E. faecalis* (eFluor⁺GFPhi) and non-replicating *E. faecalis* (eFluor⁺GFPlo) (Fig 4A). At 2 hpi, <0.5% of infected macrophages contained replicating bacteria (GFPhi macrophages), compared to 1–3% and 3–5% at 6 and 20 hpi, respectively (Fig 4A–4B). Confocal microscopy confirmed that GFPhi macrophages contained clusters of replicating *E. faecalis* with a diluted eFluor signal, while GFPlo macrophages contained lower numbers of non-replicating eFluor⁺ *E. faecalis* (Fig 4C). Quantitative PCR (qPCR) showed that the shift from the extracellular to intracellular environment led to an early decrease in Fsr QS-regulated genes compared to extracellular *E. faecalis*, likely due to a shift into a microenvironment with little to no GBAP autoinducer peptide. At 6 hpi, replicating *E. faecalis* within GFPhi macrophages significantly upregulated the *fsrABDC* locus genes and its known regulon compared to non-replicating intracellular bacteria within GFPlo macrophages at the same timepoint, approaching extracellular expression levels (Figs 4D and S10). By 20 hpi, expression levels were similar between replicating (GFPhi population) and non-replicating (GFPlo population) intracellular *E. faecalis* (Fig 4E). These results suggest that intracellular *E. faecalis* activates the Fsr QS system and its regulon following early replication within macrophages, likely in response to increased intracellular density, initiating a regulatory program that ultimately promotes the intracellular-to-extracellular transition.

### Loss of GelE proteolytic activity delays egress from macrophages and promotes intracellular accumulation of *E. faecalis* in macrophages

We hypothesized that GelE secreted by replicating intracellular bacteria promotes host cell lysis and bacterial release, thereby reducing the intracellular bacterial population. To test this hypothesis, we quantified the lytic release of intracellular lactate dehydrogenase (LDH) from macrophages. At 6 hpi, infection with GelE⁺ strains caused ~5% macrophage death, whereas GelE⁻ strains caused significantly lower cytotoxicity comparable to uninfected controls (Fig 5A). By 20 hpi, comparable cytotoxicity was seen in both GelE⁺ and GelE⁻ strain infections. These data suggest that intracellular GelE accelerates macrophage lysis and bacterial release into the antibiotic-containing media, eventually reducing overall intracellular GelE⁺ populations by 20 hpi.

To investigate the role of GelE in promoting egress, we performed a modified antibiotic protection assay up to 6 hpi, following which the bactericidal cocktail was replaced with vancomycin at a bacteriostatic concentration to *E. faecalis*, allowing intracellular bacteria to exit without extracellular replication or death [55] (S11A–S11C Fig). Supernatants were sampled hourly from 6-12 hpi for CFU enumeration. WT *E. faecalis* egressed rapidly, plateauing by 7 hpi, whereas Δ*gelE* egressed more slowly (Fig 5B). Complementation restored Δ*gelE* to WT egress kinetics, while protease-inactive E329A and E352A mutants remained defective (Fig 5C). The weakly GelE⁺ Δ*fsrBDC* mutant displayed an intermediate phenotype, supporting a correlation between gelatinase activity and egress efficiency. Both Δ*fsrA* and Δ*fsrABDC* surpassed WT egress levels at later time points, suggesting that additional FsrA-regulated factors may restrict intracellular replication or release. Consistent with the extracellular CFU recovered, confocal microscopy of infected Δ*gelE*-infected macrophages between 6–12 hpi showed large, highly dense intracellular bacteria encased within F-actin (Fig 5D, white and yellow

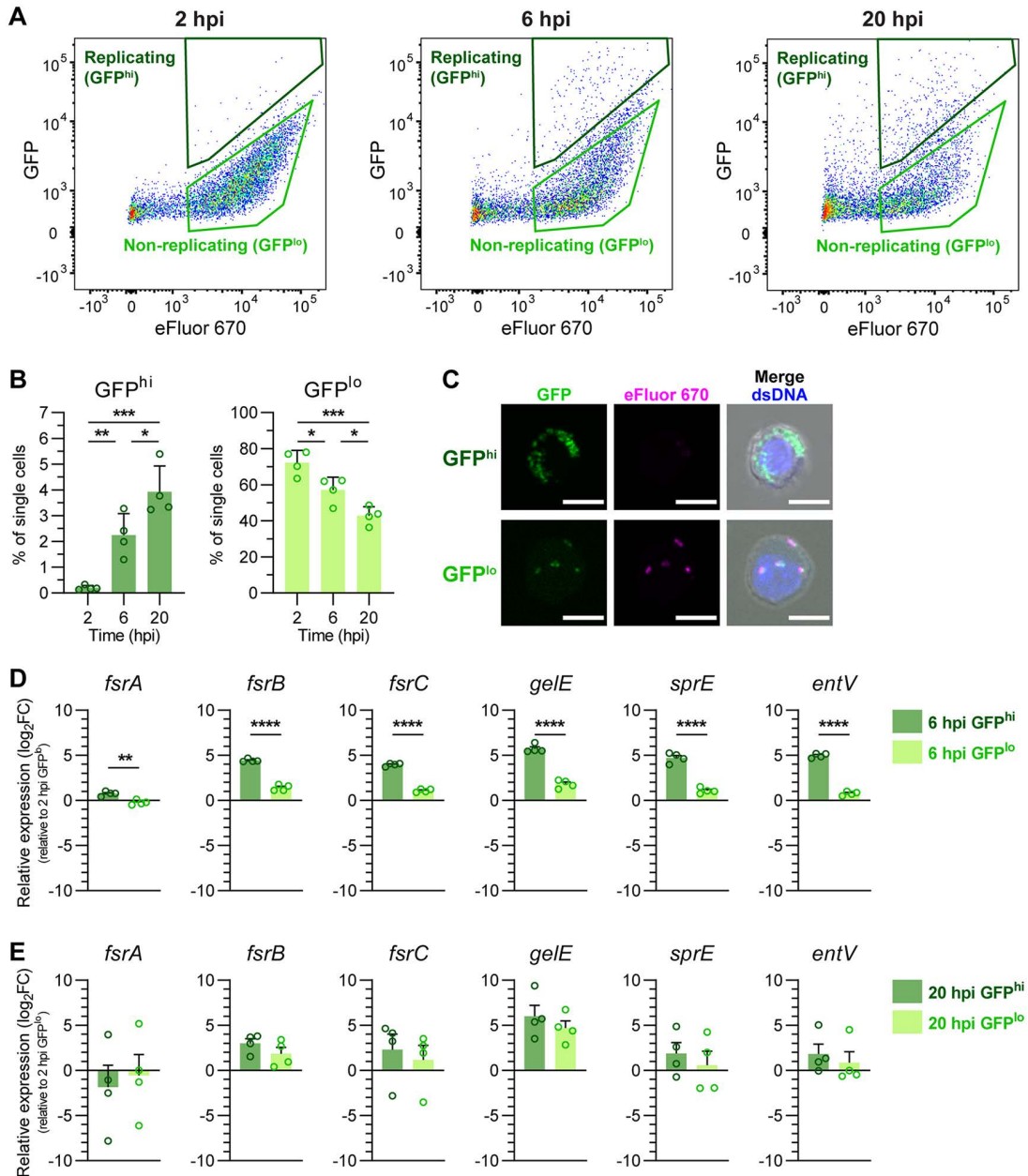

**Fig 4. Intracellularly replicating WT OG1RF induce Fsr quorum sensing at 6 hpi. (A)** Gating of eFluor+GFPhi (GFPhi) and eFluor+GFPlo (GFPlo) infected macrophages used for cell sorting at 2, 6 and 20 hpi. Representative dotplots from n = 4 are shown. **(B)** Relative quantification (%) of GFPhi and GFPlo macrophages as gated in **(A)**. Bars represent mean ± SD of n = 4. Statistical analysis was assessed using one-way ANOVA with Tukey's multiple comparisons test. **(C)** Confocal fluorescence images of sorted GFPhi and GFPlo macrophages, with intracellular GFP+eFluor670lo bacteria indicating proliferating *E. faecalis* (n = 1). Scale bar = 10 μm. **(D-E)** Relative gene expression of *fsrABDC* and its associated regulon in intracellular replicating (GFPhi) and non-replicating (GFPlo) *E. faecalis* populations at **(D)** 6 hpi and **(E)** 20 hpi, normalised by the -ΔΔCt method to the housekeeping gene *recA* and to the gene expression of the baseline 2 hpi GFPlo population. Bars represent mean ± SEM of n = 4. Statistical significance between GFPhi and GFPlo populations at each timepoint was assessed using unpaired T-test. For all graphs, only comparisons with p < 0.05 are annotated. * = p < 0.05, ** = p < 0.01, *** = p < 0.001, **** = p < 0.0001.

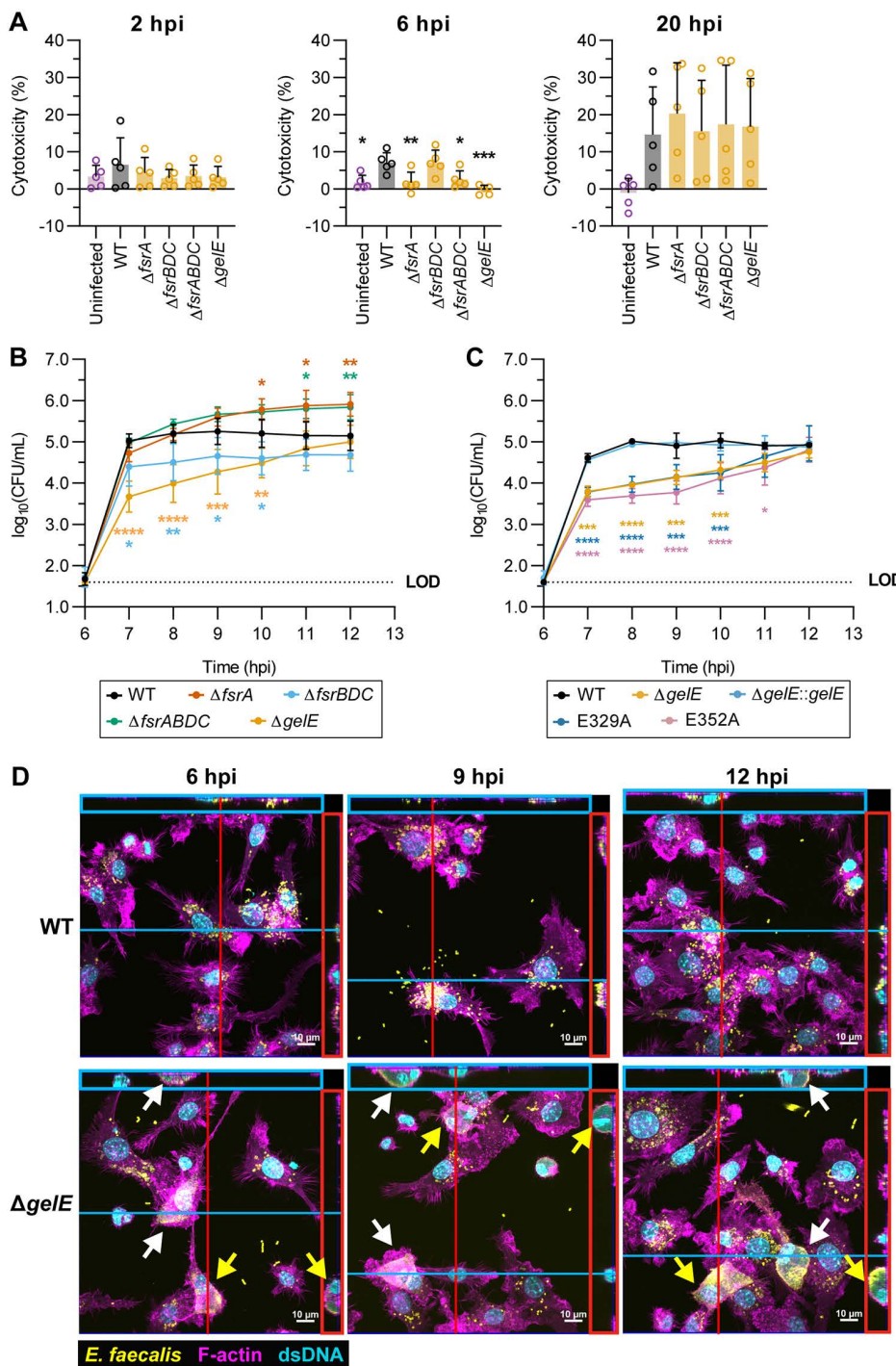

**Fig 5. Loss of GelE proteolytic activity delays egress from macrophages and promotes intracellular accumulation of *E. faecalis* in macrophages. (A)** LDH cytotoxicity quantification of culture supernatants from *E. faecalis*-infected macrophages at 2, 6 and 20 hpi using the antibiotic protection assay. Bars represent mean±SD from n = 5. Statistical significance against WT was assessed using one-way ANOVA with Dunnett's multiple comparisons test. A subset of this data comparing macrophage cytotoxicity from WT infection at 2, 6 and 20 hpi is shown in S1B Fig. **(B-C)** Quantification of extracellular egressed bacteria from culture supernatants of RAW264.7 macrophages infected with **(B)** *fsrABDC/gelE* deletion mutants and **(C)** *gelE*-complemented OG1RF strains at 6 to 12 hpi. Each data point represents mean±SD of n = 3-4. At each timepoint, statistical significance against WT was assessed using two-way ANOVA with Dunnett's multiple comparisons test. LOD = limit of detection. **(D)** Representative Z-projections of WT- and

Δ*gelE*-infected RAW264.7 macrophages at 6, 9 and 12 hpi from **(B)**, captured with confocal microscopy and stained for *Enterococcus* specific Group D antigen (yellow), dsDNA (cyan), and F-actin (magenta) (n = 2). Orthogonal Z-axis projections along the blue and red lines are shown in the adjacent colored boxes. White and yellow arrows indicate matched top view and Z-axis projections of selected macrophages with dense intracellular *E. faecalis*. For all graphs, only comparisons with p < 0.05 are annotated. * = p < 0.05, ** = p < 0.01, *** = p < 0.001, **** = p < 0.0001.

arrows), which were largely absent in WT-infected macrophages, suggesting a population of continuously replicating Δ*gelE* that is egress-defective. By contrast, WT-infected macrophages harbored smaller and more dispersed bacterial clusters, consistent with active egress and host cell lysis. To investigate whether intracellular replication and the formation of dense intracellular clusters occurred in the host cytosol, we applied a differential permeabilization method for staining cytosolic but not phagosomal bacteria in infected host cells [56–58]. Although we found large replicating intracellular clusters within the host cytosol by 6 hpi, several smaller clusters of replicating *E. faecalis* were observed to be non-cytosolic, and these were observed in both WT- and Δ*gelE*-infected macrophages (S12A Fig). These results suggest that *E. faecalis* intracellular replication is likely a heterogenous process, in which large replicating intracellular *E. faecalis* clusters eventually localize to the cytosol regardless of *gelE* presence. Transmission electron microscopy further supported the heterogenous subcellular localization of both WT and Δ*gelE* at 6 hpi in both host membrane-bound compartments and the cytosol (S12B Fig). Taken together, these data show that GelE proteolytically regulates *E. faecalis* egress from RAW264.7 macrophages, without which replicating bacteria remain as dense cytosolic populations due to delayed egress.

### Absence of GelE promotes intracellular survival of *E. faecalis* in murine wounds

Intracellular bacterial populations of traditionally extracellular pathogens can contribute to persistent and/or recurrent infections, including wound infection, bacteremia, and lung infection [59–63]. To evaluate whether GelE deficiency affects *E. faecalis* persistence *in vivo*, we compared WT and Δ*gelE* in a mouse wound infection model. Both strains established infection and persisted at similar levels at 1, 3, and 5 dpi in mono-species (Fig 6A) and competitive infections (Figs 6B and S13), showing that GelE is dispensable for wound infection at these timepoints.

We next investigated whether Δ*gelE* contributes to intracellular persistence during wound infection. At 5 dpi, dissociated wound tissues were analyzed by flow cytometry for relevant immune subpopulations [64] (S14 Fig). Both WT and Δ*gelE* infections elicited comparable innate immune infiltration at 5 dpi, with greater populations of neutrophils (CD45+ Ly6G+) and monocytes (CD45+ Ly6G- Ly6C+) compared to mock-infected wounds (S15A–S15B Fig). Intracellular *E. faecalis* were found in approximately 0.1-5% of all the cell types investigated for both WT and Δ*gelE* infection (Fig 6C). To quantify intracellular CFU, excised wound tissues were enzymatically dissociated *ex vivo* in antibiotics-containing medium (500 μg/mL each of gentamicin and penicillin G) to eliminate extracellular bacteria. The single cell suspension was then washed and lysed to release intracellular bacteria. While intracellular CFU were similar between WT and Δ*gelE* at 1 and 3 dpi, significantly more intracellular Δ*gelE* CFU were present at 5 dpi as compared to intracellular WT (Fig 6D). Parallel experiments without *ex vivo* antibiotic treatment to capture both intracellular and extracellular bacteria yielded no significant differences in total CFU between WT and Δ*gelE* (Fig 6E). Thus, although GelE does not contribute to overall fitness during wound infection, its absence promotes *E. faecalis* intracellular survival during wound infection.

### Discussion

Here, we identified the Fsr-regulated secreted metalloprotease GelE as a key regulator of *E. faecalis* intracellular dynamics. While *E. faecalis* has long been considered an extracellular pathogen, our findings reveal that it transiently adopts an intracellular lifestyle in both macrophages and wound infection-associated cells, and that GelE activity critically governs the transition between intracellular replication and extracellular dissemination. Loss of GelE activity enhances intracellular bacterial accumulation by preventing timely host cell lysis and egress (Fig 7). The frequent loss of *fsrABDC* and *gelE* among clinical isolates underscores the clinical relevance of this pathway. In a study of enterococcal infective

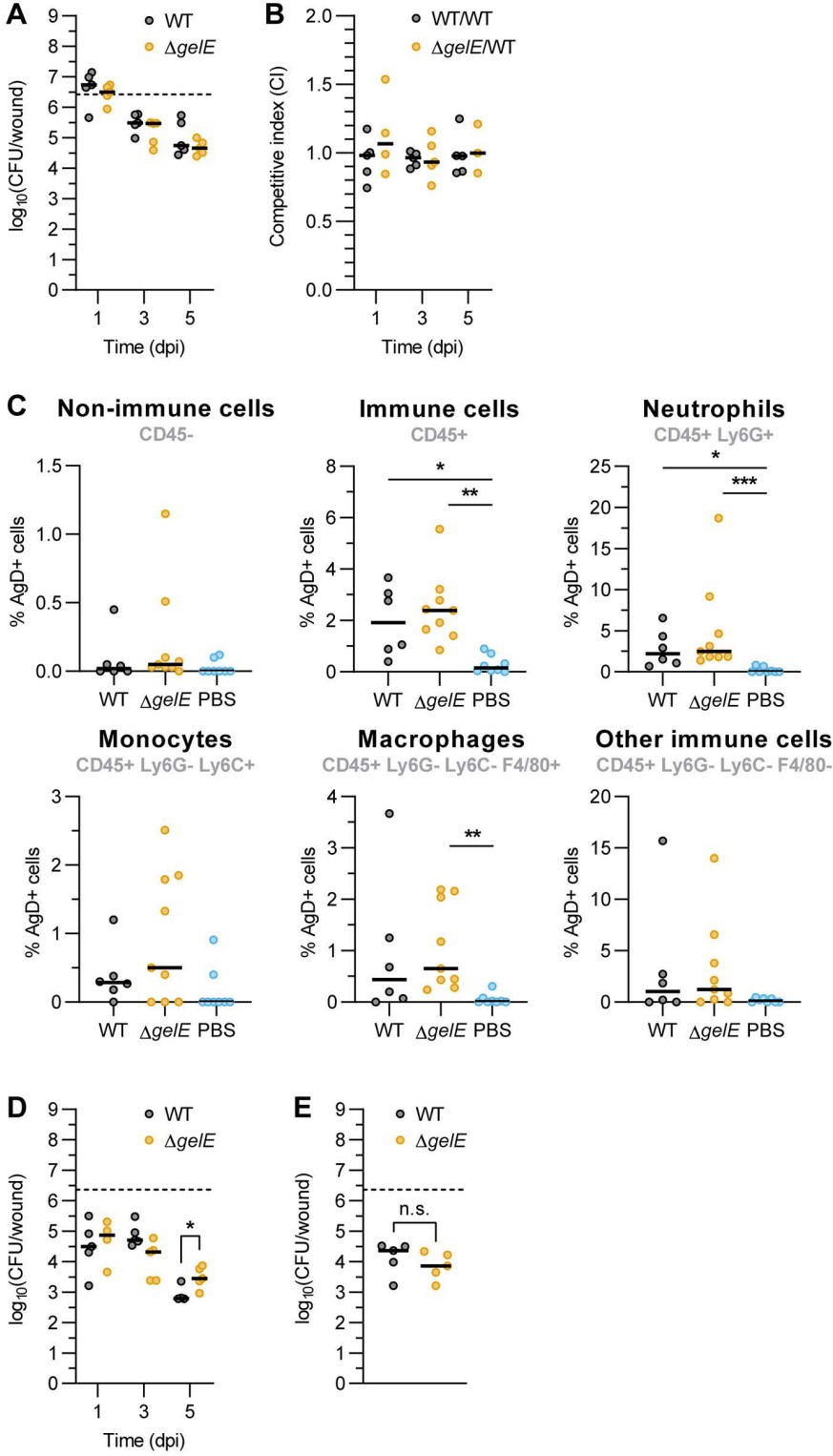

**Fig 6. Absence of GelE promotes intracellular survival of *E. faecalis* in murine wounds. (A-B)** Total CFU from wound homogenates of **(A)** mono-infection or **(B)** competitive infection at 1, 3, and 5 dpi. Median of 3-5 animals per group from one independent experiment is shown. Dotted lines show inoculum CFU for mono-infection. **(C)** Flow cytometry analysis of intracellular *E. faecalis* (stained for Group D antigen) in various immune cells at

5 dpi. Median from 6-9 animals per group from 2 independent experiments is shown. **(D-E)** Quantification of **(D)** intracellular CFU at 1, 3 and 5 dpi or **(E)** total CFU at 5 dpi from enzymatically dissociated wounds. Median from 4-5 animals per group from one independent experiment is shown. For A-D, only comparisons with p < 0.05 are annotated. At each timepoint, statistical significance between infection groups was assessed using Mann-Whitney test, except for C, which was assessed using Kruskal-Wallis test with Dunn's multiple comparisons test. * = p < 0.05, ** = p < 0.01, *** = p < 0.001. n.s. = not significant.

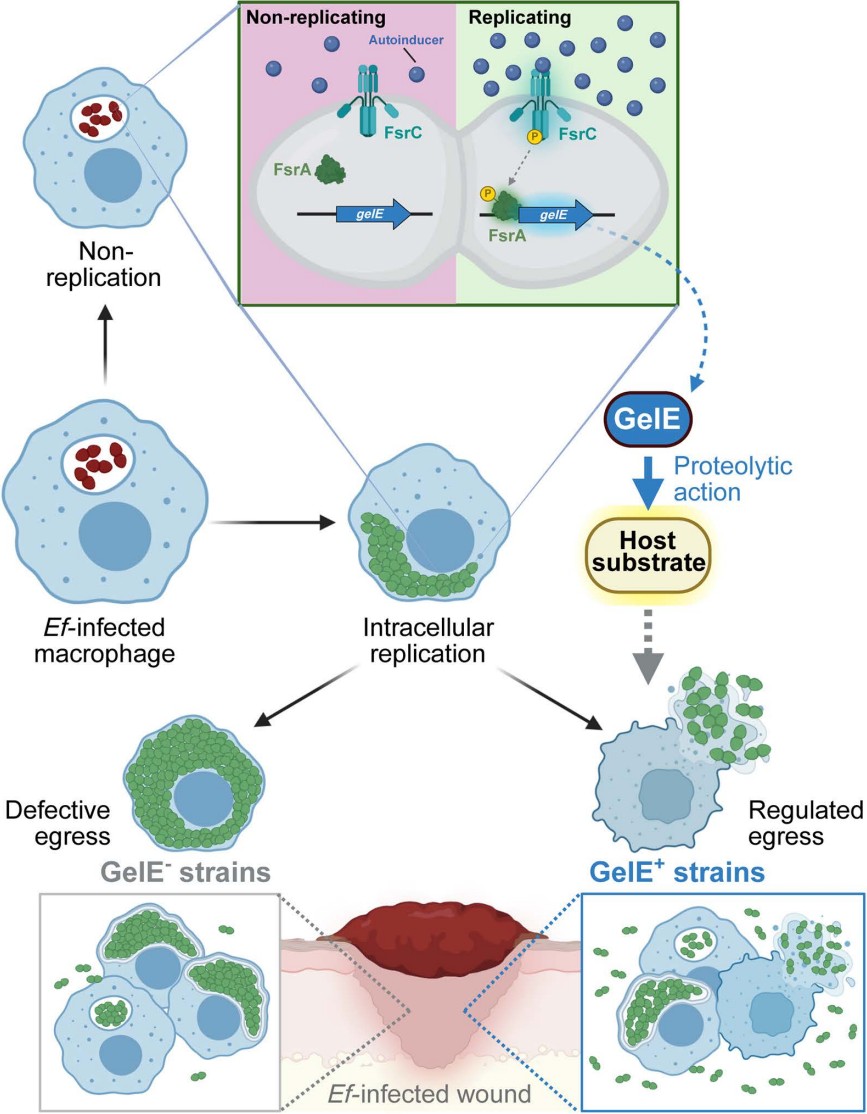

**Fig 7. Increased expression of QS-regulated GelE in replicating populations drives egress from macrophages and wound cells.** Replicating intracellular *E. faecalis* exhibit Fsr QS-mediated induction of *gelE* expression compared to non-replicating populations, promoting egress from macrophages and from wound infection-associated cells via its proteolytic action on an unknown host substrate. By contrast, absence of GelE resulted in egress-defective, dense intracellular populations. Created in BioRender. Tanoto, F. (2026) https://BioRender.com/eo3dbrt.

endocarditis, approximately 47% of *E. faecalis* isolates across two patient cohorts were *fsrABDC*- and were associated with higher disease severity and longer bacteremia duration [65]. Similarly, we found that GelE- *E. faecalis* appears better adapted for persistence and immune evasion rather than acute virulence. This evolutionary trade-off may favor intracellular survival during chronic infection or antibiotic exposure. These findings expand current understanding of *E. faecalis* pathogenesis by linking a canonical extracellular virulence factor to the control of intracellular replication [6,10,12].

The new role for GelE in mediating egress from host cells is independent of its known bacterial proteolytic targets, because neither *ace* [40], *atlA* [41], nor *entV* [42] mutants accumulate intracellular CFU as a *gelE* mutant does, suggesting that GelE may directly act on host targets to promote egress. Protease-driven egress has been previously reported in other pathogens. In *Chlamydia trachomatis*, the chlamydial protease-like activity factor (CPAF) degrades the cytoskeletal protein vimentin as well as SLC7A11, a membrane antiporter, altogether inducing ferroptosis to promote its lytic release from inclusion bodies [66,67]. Various intracellular parasitic eukaryotes, including *Plasmodium falciparum, Toxoplasma gondii* and *Trypanosoma cruzi* also utilize secreted proteases to degrade the cytoskeleton for egress [68–70]. On the other hand, *Streptococcus pyogenes* SpeB activates the pore-forming gasdermin A to induce pyroptosis [71,72], while *S. aureus* staphopain A induces host cell death through unknown effectors [73]. Similarly, GelE may cleave host factors that modulate membrane integrity or cell death pathways, potentially involving similar cytoskeletal, pore-forming or apoptotic effectors described for other pathogens. Additional cellular and molecular studies are required to clarify GelE targets within the host intracellular environment.

Loss of the *fsrABDC* and *gelE* loci is a common feature of *E. faecalis* clinical isolates. In our analysis, more than half of the *E. faecalis* wound isolates were phenotypically GelE-, caused by a genetic loss of either *fsrABDC* or both *fsrABDC* and *gelE*, but not loss of *gelE* alone. This genotype distribution is comparable with previous studies which found phenotypic GelE- strains (*fsrABDC*-*gelE*+ and *fsrABDC*-*gelE*-) in approximately 25–50% of all isolates from urine, blood, endocarditis, gut and wounds [44–47]. While the conserved 23.9 kb genomic deletion in our *fsrABDC*-*gelE*+ strains has been subject to previous investigations [45,50,51], the other conserved 27.8 kb genomic deletion encompassing *fsrABDC*-*gelE*- has been largely overlooked. How these conserved genetic deletions arise is unknown. The occurrence of conserved *fsrABDC* deletions across different sequence types supports the idea of independent adaptive events rather than clonal expansion [51]. Moreover, additional evidence from infective endocarditis suggests that these deletions may not be the result of selective pressure in the bloodstream [65]. It remains to be investigated whether other host niches or environmental reservoirs select for these mutations, but their occurrence across clinical isolates suggests they may confer an advantage in the host, such as enhanced intracellular persistence at late infection stages. Similarly, loss of the Agr QS in *S. aureus*, a homolog of the Fsr system [33], has been associated with persistent bacteremia and increased mortality [74–77]. Although we did not observe decreased virulence in the absence of *gelE* during wound infection, there was an increase in intracellular reservoirs of Δ*gelE E. faecalis* at later timepoints of infection. Future experiments incorporating even later time points or antibiotic pressure may reveal whether these intracellular populations contribute to persistence or treatment failure. On the other hand, the genetic presence of other QS systems may conversely augment intracellular persistence, as shown for two novel purine QS systems of *E. faecalis* [32] and other QS systems of Gram-negative bacteria [78,79]. Together, these observations highlight the versatile roles of QS in intracellular persistence and pathogenesis.

In summary, our findings provide new insights into the intracellular lifestyle of *E. faecalis* and reveal a QS-driven strategy that regulates its intracellular-to-extracellular transition. This work also highlights that intracellular replication and egress dynamics are heterogeneous across infected cells, which may correlate with the differential phagosome trafficking and cytosolic escape previously reported for *E. faecalis* [6–8]. Furthermore, it has been long recognized that the *fsr* and *gelE* loci are genomically heterogenous among clinical isolates, but the functional significance of this has been unclear. Our work increasingly shows how this variability has phenotypic consequences, influencing QS activity and thereby shaping intracellular persistence and egress strategies during infection. Deciphering how bacterial genotype and host

environments interact to control the intracellular-to-extracellular transition of *E. faecalis* will be critical for developing therapies that effectively clear persistent bacterial infections.

## Materials and methods

### Ethics statement

All animal infection procedures were approved and performed in accordance to the requirements of Institutional Animal Care and Use Committee in Nanyang Technological University (ARF SBS/NIEA0198Z). Murine wound infections were performed according to the approved animal use protocol (AUP) "Mechanisms of Polymicrobial Wound Infection" (AUP codes: A19061, A24066). Anonymized bacterial isolates used in this study did not require IRB approval, as described previously [49].

### Bacterial strains and culture

*Enterococcus faecalis* strain OG1RF [80] and its derivative transposon insertion mutants [39], genetic deletion mutants and other isogenic strains (S1 Table), as well as strain JH2–2 [52] and clinical wound isolates of *E. faecalis* previously collected by Timothy Mark Sebastian Barkham from patients in Tan Tock Seng Hospital, Singapore (S2 Table) were all grown statically in Brain Heart Infusion (BHI) broth (Neogen, Singapore) and BHI + 1.5% Bacto-Agar (Becton-Dickinson, USA). For all experiments, bacterial suspensions were normalized to an optical density at 600 nm ($OD_{600}$) of 0.5, corresponding to approximately $3.0 \times 10^8$ colony forming units (CFU)/mL, prior to further bacterial count normalisations. Where appropriate, we added chloramphenicol (Cm) at 10 µg/mL, rifampicin (Rif) at 25 µg/mL, spectinomycin (Spec) at 60 µg/mL (OG1RFS) or 120 µg/mL (plasmid selection), and erythromycin (Erm) at 25 µg/mL (*E. faecalis*) or 500 µg/mL (*E. coli*).

### Mammalian cell culture

RAW264.7 cells (ATCC, USA) or RAW-Blue cells (Invivogen, Singapore), a derivative of RAW264.7 cells containing a SEAP reporter plasmid, were cultured in complete Dulbecco's Modified Eagle Medium (complete DMEM; Gibco, Singapore) supplemented with 10% heat-inactivated fetal bovine serum (HI-FBS; Gibco, Singapore) at 37°C, 5% $CO_2$. 200 µg/mL Zeocin (Gibco, Singapore) was added for RAW-Blue cell line maintenance as per manufacturer's protocol.

### Antibiotic protection assay

$1 \times 10^6$ or $3 \times 10^5$ RAW-Blue cells were seeded per well of a 6-well plate or 24-well plate respectively and left to attach overnight at 37°C, 5% $CO_2$, and overnight cultures of *E. faecalis* were incubated statically at 37°C. The following day, overnight cultures were subcultured 1:10 in BHI broth to log-phase growth at 37°C, washed, and normalised to $OD_{600}$ of 0.5 in complete DMEM. Macrophages were infected with *E. faecalis* at MOI 10, centrifuged at 500 x g for 5 min to enhance macrophage-bacteria contact, and incubated for 1 h at 37°C, 5% $CO_2$. When indicated, the bacterial inoculum was pre-stained with eFluor 670 proliferation dye prior to normalization in DMEM, as described for flow cytometry. At 1 hpi, media was changed to complete DMEM with 10 µg/mL vancomycin (Sigma, USA) and 150 µg/mL gentamicin (MP Biomedicals, USA) to kill extracellular bacteria until indicated timepoints. Then, infected macrophages were washed thrice with Dulbecco's phosphate buffered saline (DPBS; Gibco, Singapore) and lysed with 1 mL 0.1% Triton X-100 (Sigma, USA) in DPBS with vigorous pipetting. Lysates were serially diluted, spotted on BHI agar, and incubated at 37°C overnight for intracellular CFU enumeration. The antibiotic susceptibility of OG1RF, JH2–2 and selected clinical wound isolates tested in this antibiotic protection assay is verified in S5B Fig.

### Molecular cloning

All plasmids were transformed and propagated in Stellar *E. coli* competent cells (TakaraBio, USA), which were cultured aerobically in LB with the appropriate antibiotics at 37°C, except for pFR213-transformed *E. coli* strains which were

cultured aerobically in BHI with 100 µg/mL erythromycin at 30°C. Plasmids were extracted using Monarch Plasmid Mini-prep Kit (New England BioLabs, Singapore) and verified by Sanger sequencing prior to transformation into *E. faecalis*. The plasmids and primers used in this study are detailed in S3 and S4 Tables respectively. All engineered strains were confirmed by whole genome sequencing to have the intended insertions/deletions with minimal off-target mutations.

Chromosomal insertions or deletions were performed by allelic exchange using the temperature sensitive shuttle plasmid pGCP213 [81] or its derivatives, pFR212 and pFR213, which were generated in this study as follows: DasherGFP was first PCR amplified from pBSU101::DasherGFP [82] using Q5 High-Fidelity DNA Polymerase (New England BioLabs, Singapore), and its ends were double-digested with SacI/SphI (New England Biolabs, USA). The insert was then ligated via T4 ligase (New England Biolabs, USA) to SacI/SphI-digested pSD15 [83], which contains full OG1RF_11778 (abbreviated to 11778) and OG1RF_11779 (abbreviated to 11779) sequences flanking the multiple cloning site for genomic insertion site for expression (GISE) [83], generating pSD15::DasherGFP. The constitutive DasherGFP promoter was then replaced via InFusion Cloning (TakaraBio, USA) with *gelE* promoter region ($P_{gelE}$; -86 bp to +121 bp, relative to +1 transcriptional start site), which was PCR amplified from WT OG1RF genomic DNA, generating pSD15::$P_{gelE}$-DasherGFP. The entire GISE cassette, which is the 11779-$P_{gelE}$-DasherGFP-11778 sequence from pSD15::$P_{gelE}$-DasherGFP, was then PCR amplified and ligated to pGCP213 via InFusion cloning, generating pFR212. Finally, a P-*pheS**  negative counterselection cassette was PCR-amplified from pRV1 [83,84] and ligated to pFR212 upstream of the GISE cassette via InFusion cloning to generate pFR213.

Chromosomal deletion strains Δ*fsrA* and Δ*fsrBDC* were generated as described previously [65]. Briefly, upstream (US) and downstream (DS) flanking regions were PCR-amplified from OG1RF genomic DNA and ligated by overlap extension PCR. The pGCP213 vector was linearized by inverse PCR, vector-insert ligation was performed via InFusion Cloning, and transformed into Stellar *E. coli* with selection on LB agar + Erm. Sequence-verified plasmids were extracted and transformed into WT OG1RF by electroporation, which were then serially passaged for allelic exchange to obtain plasmid-free chromosomal deletion strains. For the Δ*gelE* deletion mutant of OG1RFC, the chromosomal deletion was performed similarly as above, except using pFR213 instead of pGCP213, in which the full 11779-$P_{gelE}$-DasherGFP-11778 cassette in pFR213 was replaced with the US$_{gelE}$-DS$_{gelE}$ insert via InFusion cloning, generating pFR213::*gelE*_del. This allowed for a counterselection for plasmid-free clones by plating serial BHI cultures on MM9YEG agar supplemented with 10 mM *p*-chloro-phenylalanine (MM9YEG + *p*-Cl-Phe) [84]. Colonies on MM9YEG + *p*-Cl-Phe agar were then verified for Erm susceptibility, indicating loss of shuttle plasmid, prior to colony PCR. All deletion strains were verified by colony PCR and Sanger sequencing of the deletion locus using the respective sequencing primers indicated in S4 Table.

The generation of chromosomal complementation strains of Δ*gelE* (Δ*gelE*::*gelE*, E329A, E352A) have also been described previously [65]. Briefly, WT *gelE* allele including homologous flanking regions were PCR-amplified with OG1RF genomic DNA and subjected to overlap extension PCR, introducing a silent A29A (GCA > GCG) mutation to distinguish complementation *gelE* sequences from WT *gelE* allele. This complementation *gelE* insert was then ligated to pGCP213 via InFusion cloning, generating pGCP213::*gelE*. Site directed mutagenesis (SDM) by PCR was then performed on pGCP213::*gelE* using mutagenic primer pairs oFR80/oFR81 (E329A mutation) or oFR82/oFR83 (E352A mutation). The PCR products were treated with DpnI (New England Biolabs, USA), purified using the Wizard SV Gel and PCR Clean-Up Kit (Promega, USA) and re-ligated by InFusion cloning, generating pGCP213::*gelE*[E329A] and pGCP213::*gelE*[E352A] respectively. These plasmids (including pGCP213::*gelE*) were then separately electroporated into Δ*gelE* OG1RF and subjected to serial passaging for allelic exchange as described above. Correct *gelE* complementation was verified by colony PCR and Sanger sequencing of the *gelE* locus, as well as susceptibility of the complemented *gelE* locus to SacII digestion (New England Biolabs, USA).

To generate OG1RF-derived strains OG1RFC and OG1RFS for competitive infection, the $P_{gelE}$-DasherGFP cassette from pFR212 was replaced via InFusion cloning with PCR-amplified chloramphenicol resistance gene *cat* from the EfaMarTn cassette [39] or spectinomycin resistance gene *spc* from pBSU101::DasherGFP [82], generating pFR212::*cat*

and pFR212::*spc* respectively. Sequence-verified pFR212::*cat* and pFR212::*spc* plasmids were extracted and transformed into WT OG1RF by electroporation, and transformants were subjected to serial passaging for allelic exchange, similar to pGCP213 constructs. OG1RFC and OG1RFS clones were verified by colony PCR and Sanger sequencing, as well as for their susceptibility to erythromycin (indicating pFR212 plasmid loss) and their differential resistance against chloramphenicol and spectinomycin respectively (S13A Fig).

## Agar-based gelatinase assay

Up to 5 µL of overnight OG1RF cultures were spotted onto Todd-Hewitt agar (Sigma-Aldrich, USA) supplemented with 3% gelatin (Sigma-Aldrich, USA), and incubated at 37°C overnight. The following day, the plates were incubated at 4°C for 1 h, allowing any degraded gelatin around the colony to form a halo, before imaging with ProtoCOL 3 system (Synbiosis, USA). Halo area, defined as the area between the halo edge and the colony edge, was measured using Fiji 2.9.0, normalized to the known area (in mm$^2$) of the agar plate.

## Western blot

Overnight cultures of WT and mutant OG1RF strains were subcultured 1:40 in 20 mL BHI broth and incubated for up to 24 h at 37°C. At indicated timepoints, 1 mL bacterial culture was pelleted at 12,000 x g for 5 min, for which all cultures were at similar bacterial densities and growth phases (S2A Fig). Supernatants were filtered through a 0.2 µm syringe filter (Pall, USA) to remove residual bacteria, and pellets (normalized to $OD_{600}$ = 0.5 in 1 mL DPBS) were lysed in 10 mg/mL lysozyme for 1 h at 37°C. Both supernatants and cell lysates were denatured in 4X LDS Sample Buffer (Invitrogen, USA) at 100°C for 20 min, and 10 µL was loaded onto NuPAGE 4–12% Bis-Tris gel (Invitrogen, USA) and separated at 120 V, 100 min in ice-cold MOPS SDS Running Buffer (Invitrogen, USA). The gel was then transferred to a PVDF membrane using iBlot2 Gel Transfer Device (Thermo Scientific, USA), and blocked with 5% BSA (Sigma, USA) in Tris-buffered saline (Thermo Scientific, USA) + Tween-20 (Sigma, USA) (TBST) for 1 h at room temperature with shaking. The membrane was probed with 1:1000 rabbit anti-GelE polyclonal antibody (Invitrogen, Cat# PA5–117682, USA) or 1:3000 rabbit anti-SecA polyclonal antibody [85] in TBST + 1% BSA overnight at 4°C with shaking, followed by 1:5000 HRP-conjugated goat anti-rabbit IgG secondary antibody (Invitrogen, Cat# 32460, USA) in TBST + 1% BSA for 1 h at room temperature with shaking. Membranes were washed thrice with TBST after incubation with each antibody. Immunoblotted proteins were visualized with SuperSignal West Femto Maximum Sensitivity Substrate (Invitrogen, USA) in an Amersham IQ800 Imager (Cytiva, USA).

## Whole genome sequencing analysis of *E. faecalis* wound isolates

Genomic DNA from 49 wound isolates of *E. faecalis* were isolated using DNEasy Blood And Tissue Kit (Qiagen, Germany) and whole genome sequencing reads were acquired using MiSeq V3 (300 bp paired-end) (Illumina, USA) to an average coverage of 18X per genome. The reads were quality- and adapter-trimmed using bbduk (bbtools v39.17) [86] to a minimum phred score of 20 and a minimum read length of 20. Draft genomes were then *de novo* assembled using SPAdes v4.1.0 [87] with the "--careful" setting, and the assembled contigs were analyzed by blastn v2.16.0 against OG1RF nucleotide sequences of *fsrA*, *fsrB*, *fsrC* and *gelE* to identify the presence or absence of these genes. The blastn results also indicated the start and end residues of any sequence match to pinpoint the deletion junction for partial matches. To identify sequence type, trimmed reads were directly analyzed by SRST2 v0.2.0 [88] with *E. faecalis* PubMLST database [89].

To compare the genomic locus surrounding *fsrABDC/gelE*, contigs from SPAdes were scaffolded to the V583 reference genome (Accession ID: NC_004668.1) using the scaffold function of RagTag v2.1.0 [90], and were passed as queries to blastn v2.16.0 against the genomic region corresponding to EF1814-EF1844 from the V583 reference genome. Matching query start and end positions were used to extract the genomic locus from the scaffolded contig of each strain, as well as published OG1RF reference genome (Accession ID: NC_017316.1) and JH2–2 genomic scaffold (Accession

ID: KI518257.1), using samtools faidx v1.21 [91], which was subsequently annotated by prokka v1.14.6 [92] using V583 GenBank annotation (Accession ID: NC_004668.1) as reference via the "--proteins" option, and clustered and visualized for gene cluster similarity by clinker v0.0.31 [93].

### Flow cytometry of replicating intracellular bacteria in macrophage lysates

Detection of intracellular bacteria replication using eFluor 670 amine-reactive proliferation dye was adapted from Flannagan and Heinrichs [94]. Log-phase *E. faecalis* cultures in BHI were pelleted at 12,000 x g for 5 min, washed thrice with DPBS, and stained with 10 µM eFluor 670 (eBioscience, Singapore) in DPBS, at room temperature for 30 min with shaking (200 rpm). eFluor 670-stained bacteria were then washed once with BHI broth to quench unreacted eFluor 670, and once with DPBS to wash off excess media. Bacteria were then normalized to MOI 10 in complete DMEM and used for infecting macrophages as described for the antibiotic protection assay.

At indicated infection timepoints, 600 µL of intracellular lysates were aliquoted, kept on ice, and blocked for 15 min in DPBS with 2% BSA. Intracellular *E. faecalis* was labelled with 1:500 rabbit anti-Streptococcus Group D polyclonal antibody (anti-AgD; American Research Products, Cat# 12-6231D, USA) for 15 min at room temperature, followed by 1:1000 Alexa Fluor 488-conjugated goat anti-Rabbit IgG (Invitrogen, Cat# A11034, USA) for 15 min at room temperature. Washing with staining buffer was done after every immunolabelling step. Samples were then resuspended in 200 µL staining buffer and kept on ice until analysis using BD C6 Accuri flow cytometer (Becton Dickinson, USA).

### Cell sorting of infected macrophages

Prior to infection, WT OG1RF was transformed with pBSU101::DasherGFP (S3 Table, hereby referred to as pDasher OG1RF) for the constitutive expression of GFP. pDasher OG1RF was maintained in BHI agar supplemented with 120 µg/mL spectinomycin and was freshly transformed for every independent replicate. Macrophages were infected using the antibiotic protection assay with proliferation dye staining, in which RAW-Blue cells were seeded at 1 x 10$^6$ cells/well in four to eight 6-well plates per timepoint and pDasher OG1RF were inoculated and subcultured in BHI broth + 120 µg/mL spectinomycin. eFluor-stained pDasher OG1RF were normalised to MOI 10 in DMEM + 10% FBS with no spectinomycin, and macrophages were infected as described in the antibiotic protection assay. At 2, 6 and 20 hpi, the antibiotic media was aspirated, and the infected macrophages were gently scraped and collected in ice-cold 2% BSA in DPBS. Collected macrophages were pelleted at 500 x g, 4°C for 5 min, resuspended in 1–3 mL ice-cold 2% BSA in DPBS, and sorted into eFluor$^+$GFP$^{hi}$ and eFluor$^+$GFP$^{lo}$ populations in a cooled FACSFusion Cell Sorter (Becton-Dickinson, USA). Sorted cell suspensions were pelleted and resuspended in 800 µL ice-cold TRIZol reagent (Invitrogen, USA) to be transferred to Lysing Matrix B tubes (MP Biomedicals, USA). In parallel, *E. faecalis* (10$^7$ CFU inoculum) was separately grown without macrophages in DMEM + 10% FBS for 6 h, and were harvested and resuspended in Trizol without cell sorting. All Lysing Matrix B tubes were homogenised in a FastPrep-24 Bead Beating System (MP Biomedicals, USA) for one round at 6.0 m/s, 40 s and stored at -80°C prior to RNA extraction. To image sorted cell populations, approximately 10,000 eFluor$^+$GFP$^{hi}$ and eFluor$^+$GFP$^{lo}$ cells were each sorted, fixed in 4% paraformaldehyde (PFA; Sigma, USA) in DPBS for 15 min at room temperature, and resuspended in 1:1000 Hoechst 33342 (Invitrogen, Cat# H1399, USA) in PBS for 20 min at room temperature. Cells were then resuspended in 5 µL ProLong Glass Antifade Mountant (Invitrogen, Cat# P36980, USA) and spotted onto glass slides. Microscopy images were acquired at NTU Optical Bio-Imaging Centre (NOBIC) imaging facility at SCELSE using the LSM 780 confocal laser scanning microscope (Carl Zeiss, Germany) equipped with 405 nm, 488 nm, 561 nm, and 633 nm lasers.

### Total RNA extraction and quantitative PCR (qPCR)

Frozen TRIZol samples were thawed on ice, and RNA was extracted by phenol-chloroform extraction followed by purification of the aqueous phase using RNeasy Mini Kit with on-column DNAse I digestion (Qiagen, USA) following manufacturer

protocols. The quantity and quality of extracted RNA were assessed by Qubit RNA Broad Range and/or High Sensitivity kits (Invitrogen, USA) and RNA ScreenTape (Agilent, USA) respectively. Approximately 250 ng of RNA from each sample was reverse transcribed using SuperScript III First-Strand Synthesis SuperMix (Invitrogen, USA) with random hexamers as primers to capture both host and bacterial RNA, and the resulting cDNA was diluted 1:10 for use as template in gene expression analysis by qPCR. qPCR primer efficiency was determined by performing qPCRs on 10-fold serial dilutions of pooled cDNA (1:2–1:2000) and calculating efficiency (in %) using StepOne software v2.3 (Applied Biosystems, USA). Primer pairs with efficiency 90–100% were selected for subsequent qPCR experiments. All qPCR reactions were performed in three technical replicates, including no-template controls, using KAPA SYBR FAST qPCR Kit (Roche, USA) and StepOnePlus Real-Time PCR System (Applied Biosystems, USA), according to manufacturer protocols. Data were analyzed using the -ΔΔCt method and presented as $\log_2$ fold change ($\log_2$FC) relative to the control group (2 hpi GFP$^{lo}$) and to the *recA* housekeeping gene. The primers used for qPCR are listed in S4 Table.

### Lactate dehydrogenase cytotoxicity assay

To assess host cell cytotoxicity, supernatants from infected and uninfected macrophages were collected in 4 technical replicates at 2, 6 and 20 hpi following antibiotic protection assay, and subjected to the Cytotoxicity Detection Kit (LDH) (Roche, USA) following manufacturer instructions. Media-only controls and 100% dead controls (uninfected cells that were fully lysed using 0.1% Triton X-100, 30 min before each timepoint) were included as 0% and 100% cytotoxicity controls. Cytotoxicity was calculated as follows:

$$\% \text{ cytotoxicity} = \frac{A_S - A_0}{A_{max} - A_0} \times 100\%$$

in which $A_S$ represents the average absorbance at 490 nm ($A_{490}$) of 4 technical replicates, $A_0$ represents the average $A_{490}$ from media-only controls (0% cytotoxicity), and $A_{max}$ represents average $A_{490}$ of 100% dead controls (100% cytotoxicity).

### Modified antibiotic protection assay for egress measurements

To determine the minimum inhibitory concentration of vancomycin for OG1RF, 2 µL of overnight cultures of OG1RF strains were inoculated to 200 µL of colorless DMEM (Gibco, Singapore) supplemented with 10% HI-FBS and indicated concentrations of vancomycin in a 96-well plate and incubated overnight at 37°C, 5% $CO_2$. Bacterial growth after 24 h of incubation was measured as absorbance at 600 nm ($Abs_{600}$) using a Tecan Infinite M200 microplate reader (Tecan, Switzerland).

For egress measurements, RAW-Blue macrophages were infected at MOI 10 in an antibiotic protection assay. At 6 hpi, the media was sampled for CFU enumeration to ensure that no extracellular bacteria were present at the start of the measurements. The cells were then washed thrice with DPBS, the media was replaced with complete DMEM supplemented with 5 µg/mL vancomycin, and then re-incubated at 37°C, 5% $CO_2$ for up to 12 hpi. 100 µL of the media was sampled hourly for serial dilution and CFU quantification on BHI agar, and replaced with 100 µL of fresh complete DMEM + 5 µg/mL vancomycin.

For *E. faecalis*-only bacteriostatic controls, 1 µL of MOI 10/mL-normalised *E. faecalis* cultures (~1 x 10$^4$ CFU) were inoculated onto 2 mL complete DMEM + 5 µg/mL vancomycin in 6-well plates, and incubated at 37°C, 5% $CO_2$ for up to 6 h. These cultures were sampled hourly as described above to determine viable bacteria CFU in the bacteriostatic media.

### Confocal microscopy of infected macrophages

To visualise infected macrophages, RAW-Blue cells were seeded on glass coverslips in 6-well plates (5 x 10$^5$ cells/well), and antibiotic protection assay at an MOI of 10 was performed. For visualisation of replicating bacteria, macrophages were infected with *E. faecalis* pre-stained with eFluor 670 as described for flow cytometry. At 20 hpi, infected

macrophages were washed thrice with DPBS and fixed with 4% PFA for 15 min at room temperature. For egress experiments, at 6, 9 and 12 hpi, infected macrophages were spun down at 500 x g for 5 min to sediment any extracellular bacteria. Then, half of the culture media was carefully removed from the top and replaced with an equivalent volume of 8% PFA (final concentration = 4%) for 15 min at room temperature to fix the cells and extracellular bacteria. Subsequently, for all experiments, fixed cells were washed thrice with DPBS and permeabilized with 0.1% Triton X-100 for 20 min at room temperature. Samples were then blocked 2% BSA in DPBS (staining buffer) and immunolabelled with 1:500 rabbit anti-AgD in staining buffer overnight at 4°C, followed by 1:1000 Alexa Fluor 488-conjugated goat anti-Rabbit IgG for 1 h at room temperature. All samples were then stained with 1:1000 phalloidin-Alexa Fluor 568 (Invitrogen, Cat# A12380, USA) and 1 µg/mL Hoechst 33342 for 20 min at room temperature before mounting on glass slides using ProLong Glass Antifade Mountant. DPBS washes were done between every staining step. Confocal microscopy was performed at NTU Optical Bio-Imaging Centre (NOBIC) imaging facility at SCELSE using an LSM780 laser scanning confocal microscope (Carl Zeiss, Germany). Representative images are shown as maximum intensity projections of Z-stack acquisitions (0.32 µm sections) with orthologous Z-stack projections shown in blue or red adjacent boxes.

Cytosolic staining of intracellular *E. faecalis* was performed using a differential permeabilization method previously described for cytosolic *S. pyogenes* and *Salmonella* [57,56]. Briefly, following antibiotic protection assay at 2, 4 and 6 hpi, cells were washed and equilibrated ice-cold KHM buffer, before being permeabilized with 50 µg/mL digitonin for exactly 5 min on ice. Subsequently, cells were washed once with ice-cold KHM buffer and once with DPBS, and then incubated with 1:500 rabbit anti-AgD in DPBS + 10% FBS for 30 min at room temperature. Cells were then fixed with 4% PFA and stained with 1:1000 Alexa Fluor 488-conjugated anti-Rabbit IgG (Invitrogen, Cat# A11034, USA) for 1 h at room temperature. To stain for total bacteria, these cells were permeabilized with 0.1% saponin (Sigma, USA) + 10% FBS in DPBS (hereby called permeabilization buffer) for 15 min, then stained with 1:500 anti-AgD in permeabilization buffer overnight at 4°C, and subsequently with 1:1000 Alexa Fluor 568-conjugated goat anti-Rabbit IgG (Invitrogen, Cat# A11011, USA) in permeabilization buffer for 1 h at room temperature. Cells were then counterstained with 1 µg/mL Hoechst 33342 in permeabilization buffer for 20 min at room temperature before mounting onto glass slides. Cells were washed with DPBS (cytosolic bacteria staining) or permeabilization buffer (total bacteria staining) between staining steps. Confocal microscopy and Z-stack acquisition was performed as described previously. RAW-Blue cells infected with PFA-fixed *E. faecalis* were used as non-cytosolic controls.

## Transmission electron microscopy

RAW264.7 cells were seeded on glass coverslips in 6-well plates (1 x 10^6 cells/well), and antibiotic protection assay at an MOI of 10 was performed. At 6 hpi, infected macrophages were washed thrice with PBS and fixed with 2.5% glutaraldehyde and 2% PFA for 1 hour at room temperature. Cells were scraped, pelleted, and washed with PBS. Pelleted cells were then post-fixed, stained and dehydrated as described previously [95], except that uranyl acetate staining was done at 4°C overnight and dehydration by the graded ethanol series was done as follows: 2 x 50%, 70%, 90%, 2 x 100% ethanol for 10 min each, then followed by 100% ethanol for 30 min and 2 x propylene oxide for 10 min. Cells were infiltrated with Epon:propylene oxide (1:1) for 1 h, followed by infiltration with fresh 100% Epon overnight, and then embedded in resin-filled gelatin capsules and polymerized in oven at 60°C for 48 h. 50 nm thin sections were cut with Leica EMFC7 ultramicrotome and diamond knife (DiATOME) and collected onto Cu 100-hexagonal EM grid coated with Formvar plastic support film. Sections were imaged with Morgagni TEM (FEI, Netherlands) operating at 80 kV acceleration voltage equipped with Veleta side-mounted CCD camera controlled by iTEM camera software (Olympus-Soft Imaging Solutions) at the Electron Microscopy Facility (PFMU), Faculty of Medicine, University of Geneva.

## Murine excisional wound infection model

Murine wound infection was performed as described previously [48,96]. Briefly, overnight cultures of OG1RF, OG1RFC or OG1RFS strains were washed thrice with PBS and normalised to ~2.6 x 10^8 CFU/mL in PBS. For mixed species infection,

the strains were mixed 1:1 (equivalent to ~2 x 10$^8$ CFU/mL per strain) prior to infection. 7–10-week-old C57BL/6J male mice were anaesthetised, their dorsal hair trimmed by shaving and Nair depilatory cream application (Church & Dwight Co. Inc., USA), and their skin disinfected. Then, wounds were created using a 6-mm biopsy punch (Integra Miltex, USA), and 10 µL of the bacteria suspension (approximately 2.6-4 x 10$^6$ CFU) was applied and sealed with a clear wound dressing (Tegaderm 3M, USA). Mice were then euthanized at 1, 3 or 5 dpi, and wounds were excised into sterile DPBS in Lysing Matrix M (MP Biomedicals, USA). Wound were homogenized using FastPrep-24 Bead Beating System (MP Biomedicals, USA) in 5 rounds of 4.0 m/s, 20 s each. Wound homogenates were serially diluted in DPBS and plated onto BHI agar with 25 µg/mL rifampicin (OGIRF and OG1RFC + OG1RFS total counts), 60 µg/mL spectinomycin (OG1RFS) and 10 µg/mL chloramphenicol (OG1RFC) for output CFU enumeration. The competitive index (CI) of mixed-species infection was calculated as $\frac{OG1RFC_{output}/OG1RFS_{output}}{OG1RFC_{input}/OG1RFS_{input}}$.

To quantify intracellular bacteria in infected mice wounds, wound sections were excised at 1, 3 or 5 dpi and transferred into 24-well plates containing FBS-free DMEM supplemented with 500 µg/mL gentamicin and 500 µg/mL penicillin G. 250 µg/mL Liberase TL (Merck, USA) was added to this antibiotic-containing DMEM and incubated for 1 h at 37°C, 5% $CO_2$ with occasional shaking to dissociate wound cells and kill extracellular bacteria. Complete DMEM was added to stop tissue dissociation and wounds were flushed to collect the wound cell suspension. Cells were pelleted at 1000 x g, 5 min, 4°C and resuspended in 1 mL 10% FBS in DPBS for washing. Cells were pelleted once more and then lysed with 0.1% Triton X-100 in DPBS for 30 min to release the intracellular bacteria. Lysates were serially diluted and plated on BHI agar + 25 µg/mL rifampicin for CFU enumeration. For total CFU quantification using this enzymatic dissociation method, antibiotics were excluded during the tissue harvest and dissociation steps. 20 µL of the cell suspension was then transferred into 180 µL of 0.1% Triton X-100 in DPBS (10X dilution) for 30 min to release any intracellular bacteria, followed by serial dilution in PBS and plating on BHI agar + 25 µg/mL rifampicin for CFU enumeration.

To analyze the host cell profile of infected mice wounds at 5 dpi, wound cells dissociated with Liberase TL in antibiotic-containing DMEM were resuspended in 200 µL staining buffer (2% BSA in DPBS). 50 µL of each cell suspension was aliquoted and blocked with 1:50 TruStain FcX PLUS anti-mouse CD16/32 antibody (BioLegend, Cat# 156604, USA) for 30 min on ice, followed by staining with 50 µL antibody cocktail of BV510 anti-mouse CD45 (BioLegend, Cat# 103138, USA), APC anti-mouse Ly6G (BioLegend, Cat# 127614, USA), BV421 anti-mouse Ly6C (BioLegend, Cat# 128031, USA), PE anti-mouse/human CD11b (BioLegend, Cat# 101208, USA), and BV711 anti-mouse F4/80 (BioLegend, Cat# 123147, USA), each diluted to a final concentration of 1:100 in staining buffer, for 1 h on ice. The cells were then washed once with cold staining buffer, fixed with 4% PFA for 20 min at room temperature, and then stored in staining buffer overnight at 4°C. The following day, cells were permeabilized for intracellular staining by washing thrice with Intracellular Staining Perme-abilization Wash Buffer (BioLegend, Cat# 421002, USA; hereby called permeabilization buffer). Then, cells were stained with 1:500 rabbit anti-AgD diluted in permeabilization buffer for 30 min at room temperature, washed with permeabilization buffer once, and stained with 1:1000 Alexa Fluor 488-conjugated goat anti-rabbit IgG diluted in permeabilization buffer for 30 min at room temperature. Cells were then resuspended in 500 µL staining buffer, and 400 µL of the cell suspension was aliquoted and mixed with 100 µL AccuCheck Counting Beads (Invitrogen, Cat# PCB100, USA) for analysis using BD LSRFortessa X-20 flow cytometer (Becton Dickinson, USA). Flow cytometry compensation was achieved using the AbC Total Antibody Compensation Bead Kit (Invitrogen, USA) following manufacturer protocols, and fluorescence-minus-one (FMO) controls were included to set gating thresholds. AgD$^+$ events were gated from the histogram using AgD-stained, mock-infected wound cells as AgD$^-$ gating threshold (S14 Fig).

## Data and statistical analysis

Graphs and statistical analyses were performed in GraphPad Prism 9.01 (GraphPad, USA). For flow cytometry data, visualization and quantification of populations were done using FlowJo v10.8.0 (Becton Dickinson, USA). Error bars were represented as means ± standard deviation (SD) or means ± standard error of mean (SEM), as indicated in the respective

figure captions. Statistical tests were indicated in individual figure captions, with statistical significance represented as follows: * $p < 0.05$, ** $p < 0.01$, *** $p < 0.001$, **** $p < 0.0001$.

## Supporting information

**S1 Fig. Intracellular CFU of *E. faecalis* in RAW264.7 macrophages peaks at 6 hpi for WT OG1RF and is comparable at 2 hpi across OG1RF strains tested.** (A) Antibiotic killing kinetics under conditions used for extracellular killing of *E. faecalis* in the antibiotic protection assay. *E. faecalis* OG1RF log-phase cultures inoculated at $10^7$ (MOI 10 equivalent; purple lines) or $10^4$ CFU (pink lines) are not recovered after ≥ 1 h of vancomycin (10 μg/mL) + gentamicin (150 μg/mL) treatment in DMEM + 10% FBS, in the absence of host cells. Data points are mean ± SD of n = 3–4. LOD = limit of detection. (B) Cytotoxicity measurements of infected macrophages during the antibiotic protection assay. WT OG1RF infection data, shown here for ease of presentation, are also shown in Fig 5A in the experimental context (comparison to mutant strains) they were originally collected from. Statistical significance between timepoints was assessed by one-way ANOVA with Tukey's multiple comparisons test (n = 5). n.s. = not significant. (C) Intracellular CFU in RAW264.7 macrophages infected with WT OG1RF at 2, 6, and 20 hpi using the antibiotic protection assay. Counts from the same biological replicate are connected by dotted lines (n = 8–12). (D) Fold-change analysis of intracellular CFU quantified in (C), normalized to the intracellular CFU at 2 hpi for each biological replicate. Dotted line indicates baseline CFU at 2 hpi (fold-change CFU = 1.0). Statistical significance between 6 and 20 hpi was assessed by unpaired T-test (n = 8–12). (E-F, H) Intracellular CFU in RAW264.7 macrophages infected with OG1RF-derived (E) mutants of genes implicated in intracellular persistence (blue), virulence (orange), or GelE proteolytic targets (pink) (n = 5–18), (F) genetic deletion mutants of the *fsr* operon and *gelE* as well as a transposon insertion mutant of *sprE* (n = 5–9) or (H) *gelE*-complemented strains at 2 hpi (n = 4). Statistical significance of each strain against WT was assessed using one-way ANOVA with Dunnett's multiple comparisons test. * = $p < 0.05$. (G) Gelatinase activity of *sprE*::Tn (representative of n = 3) compared to WT (n = 1) on Todd-Hewitt agar + 3% gelatin at 24 h. Halo formation (white dashed line) indicates gelatinase activity.
(TIF)

**S2 Fig. Proteolytic mutants E329A and E352A show defects in extracellular secretion and autocatalytic processing of GelE.** (A) Growth curve of OG1RF deletion and complementation strains in BHI broth for 24 h following 1:40 subculture from overnight cultures, plotted as mean ± SD of n = 2 with 4–5 technical replicates each. Timepoints analyzed for GelE production are marked in grey dashed lines. (B) Detection of intracellular GelE from *E. faecalis* cell lysates at 4, 12 and 24 h (harvested together with supernatants in Fig 1G and 1J). The membrane protein SecA was included as loading controls. White arrowheads = Pro-GelE (~55 kDa). < 50 kDa bands are also observed in Δ*gelE* cell lysates and are therefore likely non-specific. Images from n = 1 are shown.
(TIF)

**S3 Fig. Catalytic E329 and zinc-coordinating E352 in the GelE active site are structurally conserved in GelE and other M4-family metalloproteases.** Identification of putative key GelE active site residues E329 and E352 for proteolytic activity, based on structural homology of AlphaFold2-predicted GelE structure (pink) to other M4 family zinc metalloproteases ProA (PDB 6YA1; white) and vibriolysin MCP-02 (PDB 3NQX; yellow). Based on previous studies, only mutations in residues E346 and E369 of MCP-02 (homologous to E329 and E352 of GelE respectively) produced stable, non-proteolytic proteases (Gao et al, 2010). Protein structures were aligned in WinCoot v1.1.18 using the Secondary Structure Matching (SSM) Superpose function and visualized in PyMOL v2.5.3.
(TIF)

**S4 Fig. *fsrABDC*-*gelE*+ and *fsrABDC*-*gelE*- wound isolates of *E. faecalis* show distinct and conserved genomic deletions. (A)** Blastn query matches of contigs from all *fsrABDC⁻gelE⁺* strains to *fsrA, fsrBDC, fsrC* or *gelE*. The "Start"

and "End" columns indicate nucleotide positions of matches in the respective genes, while dashes indicate no matches, as visually represented by the schematic diagram. Sequence type (ST) of each strain is shown, with asterisks (*) indicating uncertainties in SRST2 sequence type calling. **(B)** Alignment of the genomic region corresponding to EF1814-EF1844 of the V583 reference genome for *E. faecalis* wound isolates. OG1RF and JH2–2 were included as reference strains. Genes with sequence identity > 30% are connected by shaded lines, colored by identity (%) as indicated in the legend. Colored arrows indicate *fsrA, fsrBDC, gelE* and *sprE* operons, as well as other genes flanking genomic deletion regions. Inset shows a diagrammatic summary of the conserved genomic deletions or insertions observed. The corresponding GelE producing phenotype for each strain (as reported in Fig 2A) are shown on the left. Strains highlighted in red are further tested by *in vitro* antibiotic protection assay (Fig 2B and 2D and S5).
(TIF)

**S5 Fig. Selected wound isolates of *E. faecalis* exhibit gelatinase activity expected from their genotype and are phenotypically gentamicin-susceptible for the antibiotic protection assay.** (A) Gelatinase activity of *E. faecalis* wound isolates and JH2–2 on Todd-Hewitt agar + 3% gelatin at 24 h. Halo formation (white dashed line) indicates gelatinase activity. Genotypes of wound isolates are indicated on the left. Representative images of n = 3 are shown. (B) Validation of bactericidal activity of vancomycin (10 µg/mL) + gentamicin (150 µg/mL) on log-phase cultures of *E. faecalis* wound isolates and laboratory strains (JH2–2; OG1RF WT, Δ*fsrABDC*, Δ*gelE*). Representative images from post-treatment serial dilutions are shown (n = 3). Left column = untreated controls.
(TIF)

**S6 Fig. Clinical *fsrABDC*-*gelE*+ strain EF_1008 may exhibit low-level Fsr-independent expression of gelatinase.** Detection of GelE secretion in culture supernatants of WT (*fsrABDC*+*gelE*+ control), Δ*fsrABDC* (*fsrABDC*-*gelE*+ control), Δ*gelE* (*gelE*- control), and selected clinical *fsrABDC*-*gelE*+ strains (shown in Figs 2 and S5) at 4, 12 and 24 h by Western blotting. The same blot was imaged twice at low exposure time (30 s) and high exposure time (8 min). Images from n = 1 are shown. CTD = C-terminal domain.
(TIF)

**S7 Fig. Infection efficiency at 2 hpi is comparable between GelE+ WT and GelE- deletion strains.** Flow cytometry analysis of RAW264.7 macrophages at 2 hpi infected with GFP-expressing WT (GelE+, gray bar), Δ*fsrABDC* (GelE-, orange bar) or Δ*gelE* (GelE-, orange bar) pre-stained with eFluor 670, showing the gating strategy and proportion of eFluor 670+ macrophages (infected macrophages). Uninfected macrophages are used as gating controls. Bars represent mean ± SD of n = 3, and representative dotplots from n = 3 are shown. Statistical significance was assessed by one-way ANOVA with Tukey's multiple comparisons test. n.s. = not significant.
(TIF)

**S8 Fig. Detection of replicating intracellular *E. faecalis* from macrophage lysate by flow cytometry analysis of eFluor 670 proliferation dye.** (A) Serial dilution of eFluor 670 proliferation dye in actively replicating *E. faecalis* pDash-erGFP OG1RF cultures in colorless DMEM + 10% FBS over 6 h. Bacterial cultures were aliquoted hourly for flow cytometry analysis. Representative histograms of n = 3 are shown. (B) Gating strategy for eFluor 670 analysis of intracellular *E. faecalis* released from infected RAW264.7 lysates. *E. faecalis* stained for Group D antigen (AgD) was gated as Alexa Fluor 488+ events. Debris at the histogram edges (<$10^0$ and >$10^7$ fluorescence intensity) were gated out prior to histogram bisection for accurate quantification of eFluor+ and eFluor- events.
(TIF)

**S9 Fig. *fsr/gelE* deletion mutants of *E. faecalis* OG1RF do not exhibit growth rate differences in macrophage cell culture media.** *fsr/gelE* deletion mutants were grown in colorless DMEM + 10% FBS for 24 h and the absorbance at

600 nm (Abs600) was measured at 30 min intervals. Colorless DMEM + 10% FBS is used as a negative control. Growth curve is plotted as a mean ± SD of n = 2 with 10 technical replicates each.
(TIF)

**S10 Fig. Expression of Fsr quorum sensing is lower in intracellular populations relative to extracellular _E. faecalis_.** Relative gene expression of _fsrABDC_ and its associated regulon in intracellular _E. faecalis_ populations (2 hpi GFP$^{lo}$, 6 hpi GFP$^{lo}$, 6 hpi GFP$^{hi}$) compared to extracellular _E. faecalis_ grown for 6 h in the absence of macrophages. Relative expression is calculated by the -ΔΔCt method, normalised to the housekeeping gene _recA_ and to the gene expression of 6 h extracellular _E. faecalis_. Bars represent mean ± SEM of n = 4. Statistical significance of each population against 6 h extracellular _E. faecalis_ was assessed using one-way ANOVA with Dunnett's multiple comparisons test. Only comparisons with $p < 0.05$ are annotated. * = $p < 0.05$, ** = $p < 0.01$, *** = $p < 0.001$, **** = $p < 0.0001$.
(TIF)

**S11 Fig. 5 µg/mL vancomycin is sufficient for a bacteriostatic effect on extracellular _E. faecalis_ with minimal bactericidal effect.** (A) Determination of minimum inhibitory concentration of vancomycin by broth microdilution in colorless DMEM + 10% FBS for OG1RF WT and _fsr/gelE_ deletion mutants. Bacterial growth was measured as absorbance at 600 nm (Abs$_{600}$), represented in the heatmap as mean of n = 2. (B-C) Validation of bacteriostatic effect of 5 µg/mL vancomycin for (B) _fsr/gelE_ deletion mutants (n = 4) or (C) _gelE_-complemented strains (n = 3). Recovered bacteria CFU is expressed in log$_{10}$(CFU/mL) as mean ± SD of n = 3–4.
(TIF)

**S12 Fig. WT and Δ_gelE_ OG1RF heterogenously escape to and replicate in the cytosol.** (A) Representative confocal microscopy images of RAW264.7 macrophages infected with eFluor-stained (white) WT and Δ_gelE E. faecalis_ at 2, 4 and 6 hpi (n = 2). Samples were first weakly permeabilized by digitonin to allow staining of only cytosolic bacteria using _Enterococcus_-specific antibody (green), and then fully permeabilized, including permeabilization of intracellular compartments, for total bacteria staining by the same antibody (magenta), following by counterstaining for dsDNA (blue). 6 hpi infection with PFA-fixed WT _E. faecalis_ was used as non-proliferative, non-cytosolic controls. White and yellow arrows indicate non-cytosolic bacteria population that is non-replicating (eFluor 670$^+$) and replicating (eFluor 670$^-$) respectively. (B) Transmission electron microscopy images of intracellular WT and Δ_gelE_ at 6 hpi, showing non-cytosolic bacteria encapsulated by a single membrane (blue arrows) and likely cytosolic bacteria not encapsulated in a membrane-bound compartment (red arrows). Images from n = 1 are shown.
(TIF)

**S13 Fig. Differential CFU quantification of OG1RFC and OG1RFS strains during a competitive model of wound infection.** (A) Validation of growth selection of isogenic strains OG1RFC and OG1RFS on chloramphenicol (Cm, 10 µg/mL) and spectinomycin (Spec, 60 µg/mL) BHI plates respectively, compared to parental strain OG1RF. Rifampicin (Rif, 25 µg/mL) and erythromycin (Erm, 25 µg/mL) were used as positive and negative growth controls of antibiotic selection respectively. (B-D) CFU quantification from wound homogenates on antibiotic agars, selecting for (B) total OG1RFC + OG1RFS, (C) OG1RFC, and (D) OG1RFS. Black bars represent median of 3–5 animals per infection group from one independent experiment. Statistical significance between infection groups of the same timepoint was assessed using Mann-Whitney test. Only comparisons with $p < 0.05$ are annotated. Dotted lines show bacteria inoculum CFU.
(TIF)

**S14 Fig. Flow cytometry gating and quantification of dissociated cells from 5 dpi WT, Δ_gelE_ or mock-infected murine wounds.** Wound cells (yellow panel) as well as AccuCheck Counting Beads A and Beads B (blue panel) were gated from forward-scatter/side-scatter (FSC/SSC) dotplots. Single cells from the wound were then separately gated for CD45$^+$ (immune cells) or CD45$^-$ (non-immune cells), and CD45$^+$ cells were further separated into Ly6G$^+$ (neutrophils)

and Ly6G⁻ populations. CD45⁺ Ly6G⁻ populations were subsequently gated into Ly6C⁺ (monocytes), Ly6C⁻ F4/80⁺ (macrophages), and Ly6C⁻ F4/80⁻ (other immune cells) populations. Cell counts from each gated population were normalized to the total Bead A+Bead B count from each sample. Each immune cell subpopulation was further analyzed into AgD⁺ (infected) and AgD⁻ (uninfected) cells, with AgD⁺ threshold determined in histograms based on AgD⁻ mock-infected samples.
(TIF)

**S15 Fig. Innate immune cell profile of 5 dpi WT, Δ*gelE* or mock-infected murine wounds.** (A) Cell counts from gated populations described in S14 Fig, normalized to AccuCheck Counting Bead counts. (B) Percentage of gated populations relative to all single cells (CD45⁺/CD45⁻ population) or all CD45⁺ cells (immune cell populations). Bars represent median from 6-9 mice from two independent experiments. For each cell population, statistical significance was assessed using Kruskal-Wallis test with Dunn's multiple comparison test. *=$p<0.05$, **=$p<0.01$, ***=$p<0.001$.
(TIF)

**S1 Table. Strains used in this study.** Numbers after the gene name indicate position (in bp) within the protein coding sequence where the transposon is inserted for gene disruption.
(DOCX)

**S2 Table. Details of the wound clinical isolates tested in this study.** Asterisks (*) indicating uncertainties in sequence type (ST) calling by SRST2. + and – indicates genotype/phenotype presence and absence, respectively.
(DOCX)

**S3 Table. Plasmids used in this study.**
(DOCX)

**S4 Table. Primers used in this study.**
(DOCX)

**S1 Data. Uncropped Western blot images for Figs 1G, 1J, S2B and S6.**
(PDF)

**S2 Data. Raw values used to plot all graphs in this study.**
(XLSX)

## Acknowledgments

We thank Ronni Anderson Gonçalves da Silva, Navin Jeyabalan, Antonin André and Michelle Huiying Fan for the insightful discussions about this project. We thank Gary Dunny for providing the annotated EfaMarTn transposon library, Timothy Mark Sebastian Barkham for providing the clinical wound isolates from Tan Tock Seng Hospital, Singapore, and Lok Neng Too for assisting in the molecular cloning procedures. We also thank Chen Kaiwen and Safwah Nasuha, as well as the Flow Cytometry Laboratory for facilitating the cell sorting runs at Centre for Life Sciences, National University of Singapore. Finally, we thank Navin Jeyabalan, Logeshwari Muthualagu Natarajan and Rachel Jing Wen Tan for their assistance in performing murine wound infections, and to Mélanie Roch and Antonin André for their critical reading of this manuscript.

## Author contributions

**Conceptualization:** Frederick Reinhart Tanoto, Claudia J. Stocks, Kimberly A. Kline.

**Data curation:** Frederick Reinhart Tanoto.

**Formal analysis:** Frederick Reinhart Tanoto, Jia Hui Liew, Claudia J. Stocks, Haris Antypas.

**Funding acquisition:** Kevin Pethe, Haris Antypas, Kimberly A. Kline.

**Investigation:** Frederick Reinhart Tanoto, Jia Hui Liew, Claudia J. Stocks, Deepti Rawat, Kelvin Kian Long Chong, Haris Antypas, Kimberly A. Kline.

**Methodology:** Frederick Reinhart Tanoto, Jia Hui Liew, Claudia J. Stocks, Deepti Rawat, Kelvin Kian Long Chong, Haris Antypas, Kimberly A. Kline.

**Project administration:** Kimberly A. Kline.

**Resources:** Kimberly A. Kline.

**Supervision:** Claudia J. Stocks, Kevin Pethe, Haris Antypas, Kimberly A. Kline.

**Validation:** Frederick Reinhart Tanoto, Jia Hui Liew.

**Visualization:** Frederick Reinhart Tanoto, Jia Hui Liew, Haris Antypas.

**Writing – original draft:** Frederick Reinhart Tanoto, Claudia J. Stocks, Haris Antypas, Kimberly A. Kline.

**Writing – review & editing:** Frederick Reinhart Tanoto, Jia Hui Liew, Claudia J. Stocks, Deepti Rawat, Kelvin Kian Long Chong, Kevin Pethe, Haris Antypas, Kimberly A. Kline.

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
