## [Decision Letter · Decision Letter 0]

4 Jan 2026

Gelatinase regulates the egress of intracellular replicating populations during Enterococcus faecalis infection

PLOS Pathogens

Dear Dr. Kline,

Thank you for submitting your manuscript to PLOS Pathogens. After careful consideration, we feel that it has merit but does not fully meet PLOS Pathogens's publication criteria as it currently stands. Therefore, we invite you to submit a revised version of the manuscript that addresses the points raised during the review process.

We look forward to receiving your revised manuscript.

Kind regards,

Breck A. Duerkop

Academic Editor

PLOS Pathogens

Alice Prince

Section Editor

PLOS Pathogens

Editor-in-Chief

PLOS Pathogens

orcid.org/0000-0003-2946-9497

Editor-in-Chief

PLOS Pathogens

orcid.org/0000-0002-7699-2064

**Additional Editor Comments:**

Your manuscript has been reviewed by three experts in the field. Overall, there is strong enthusiasm for the study. All reviewers indicate that the work is important and is a new contribution to our understanding of intracellular bacterial lifestyles and Enterococcus. As you will see int he reviews, most comments can be addressed with additional textual narrative and clarification. However, there are a few points raised, mostly from reviewer 1, that may require some additional experimentation. If you feel that you are able to address these questions raised, we would look forward to receiving a revised manuscript. Thank you for the privilege of reviewing your exciting study.

**Journal Requirements:**

3) Some material included in your submission may be copyrighted. According to PLOSu2019s copyright policy, authors who use figures or other material (e.g., graphics, clipart, maps) from another author or copyright holder must demonstrate or obtain permission to publish this material under the Creative Commons Attribution 4.0 International (CC BY 4.0) License used by PLOS journals. Please closely review the details of PLOSu2019s copyright requirements here: PLOS Licenses and Copyright. If you need to request permissions from a copyright holder, you may use PLOS's Copyright Content Permission form.

Potential Copyright Issues:

i) We note that Figure 7 is created through BioRender. Please confirm that you hold a Premium account and provide a pdf copy of the CC BY 4.0 Licence as provided by BioRender. For instructions on how to generate a CC BY 4.0 license for your figure, please see the guidelines here: https://help.biorender.com/hc/en-gb/articles/21282341238045-Publishing-in-open-access-resources.

If you are using the free assets from BioRender, we are unable to publish these images as they are licenced under a stricter licence than CC BY 4.0. In this case we ask you to remove the BioRender images and replace them with open source alternatives.

See these open source resources you may use to replace images / clip-art:

- https://bioart.niaid.nih.gov/

- https://bioicons.com/

- https://healthicons.org/

- https://scidraw.io/

- https://reactome.org/icon-lib

- https://www.phylopic.org/images

- https://journals.plos.org/plosbiology/article?id=10.1371/journal.pbio.3002395

4) When completing the data availability statement of the submission form, you indicated that you will make your data available on acceptance. We strongly recommend all authors decide on a data sharing plan before acceptance, as the process can be lengthy and hold up publication timelines. Please note that, though access restrictions are acceptable now, your entire data will need to be made freely accessible if your manuscript is accepted for publication. This policy applies to all data except where public deposition would breach compliance with the protocol approved by your research ethics board. If you are unable to adhere to our open data policy, please kindly revise your statement to explain your reasoning and we will seek the editor's input on an exemption. Please be assured that, once you have provided your new statement, the assessment of your exemption will not hold up the peer review process.

5) In the online submission form, you indicated that Any additional information required to reanalyze the data reported in this paper is available upon reasonable request.. All PLOS journals now require all data underlying the findings described in their manuscript to be freely available to other researchers, either

1. In a public repository

2. Within the manuscript itself

3. Uploaded as supplementary information.

**Reviewers' Comments:**

Reviewer's Responses to Questions

**Part I - Summary**

Reviewer #1: Enterococcus faecalis is a facultative anaerobe that resides as a commensal in the human gastrointestinal tract but emerges as an opportunistic pathogen in settings such as chronic wounds. In this work, Tanoto et al. link a canonical secreted virulence factor, the metalloprotease gelatinase, and its regulator, the Fsr quorum sensing system, to the regulation of intracellular replication and egress for E. faecalis within host cells. Their work builds on prior observations that a subpopulation of E. faecalis can persists intracellularly within neutrophils in a murine wound model. Using a combination of molecular cloning, microscopy, and flow cytometry, the authors demonstrate that the Fsr system is induced during intracellular replication within macrophages, promoting GelE-dependent host cell lysis and bacterial egress. In the absence of GelE, E. faecalis accumulates as large intracellular clusters of replicating cells that are egress-defective. This finding is important as it provides a mechanistic insight and clinical relevance to the frequent loss of fsr and gelE among clinical wound isolates. Though E. faecalis is traditionally regarded as an extracellular bacterium, the authors demonstrate that E. faecalis can adopt a transient intracellular lifestyle and regulates its intracellular-to-extracellular transition. While it is unclear what host substrates are required for E. faecalis egress, this work indicates that GelE is one of the important factors in this process. The data are well-presented, rigorously analyzed, and supported by appropriate controls. A few suggestions for improvement are given in the next section.

Reviewer #2: In this paper, Tanoto et al., search for E. faecalis virulence factors that control intracellular adaptation, specifically those related to quorum sensing due to the confinement of bacteria while in host cells. The authors screened for and identified GelE as a factor that controls bacterial egress from macrophages. In a series of elegant experiments, the authors demonstrate that 1) mutation of gelE and its regulator the fsr quorum sensing system are frequently mutated in wound infections, 2) GelE expression is required for bacterial egress, and loss results in accumulation of intracellular bacteria both in vitro and in vivo, and 3) egress is controlled by quorum sensing control of GelE. These results are important in understanding the ability of E. faecalis, normally considered an extracellular bacterium, to survive and persist in macrophages, and contribute to chronic wound infections. The experiments are extraordinarily rigorous and thorough, most questions I had were addressed by experiments presented in the supplemental data. The manuscript is well written and easy to follow. Clarifications requested are noted below.

Reviewer #3: Enterococcus faecalis is a frequent cause of chronic wound infections; however, the mechanisms that enable its persistence in this niche remain poorly understood. Previous work has shown that E. faecalis can survive and replicate within host cells during wound infection, suggesting that an intracellular phase is an integral to its lifestyle. Building on this observation, the authors demonstrate that the secreted metalloprotease gelatinase (GelE), regulated by the Fsr quorum-sensing system, plays a central role in mediating this process. Fsr signaling is induced during intracellular growth and promotes GelE-dependent host cell lysis and bacterial escape, whereas loss of GelE leads to the accumulation of large intracellular bacterial clusters. The authors investigate these mechanisms using well-designed in vitro infection assays in mouse macrophages, a combination of transposon and in-frame deletion mutants, genomic and phenotypic analyses of clinical E. faecalis isolates, and a mouse wound infection model.

Overall, this is an important study that probes the mechanisms underlying intracellular survival and persistence of E. faecalis in the context of wound infections. It highlights the role of GelE and the Fsr quorum-sensing system in governing the transition between intracellular and extracellular lifestyles during infection.

The manuscript is well-written, with a clear and logical flow, and the data are presented in a way that is both accessible and easy to follow. I particularly appreciate the framing of a “transient intracellular” lifestyle for certain bacterial species—demonstrated here for E. faecalis—which helps expand the field’s view beyond the traditional dichotomy of intracellular versus extracellular organisms. I also value the development and use of sophisticated in vitro assays to rigorously interrogate the biological questions addressed in this study.

The host cell response to infection by the different strains remains an important gap in the current study, particularly given that they propose one or more host-derived factors as potential targets for degradation by GelE.

**Part II – Major Issues: Key Experiments Required for Acceptance**

Reviewer #1: Can the authors provide data or cite previous evidence to clarify whether the serine protease SprE, which is also regulated by the Fsr system, contributes to the increased intracellular survival in macrophages? It is unclear whether SprE has been sufficiently ruled out as a contributing factor to intracellular survival and replication.

Line 173-183: The authors should clarify whether the absence of fsrA and gelE, directly accounts for increased intramacrophage survival, given that all fsrABCD-gelE- strains show elevated CFU by 6 hpi. In contrast, the persistence of fsrABCD-gelE+ strains appears delayed, with elevated CFU observed only at later timepoints and to a lesser degree than the fsrABCD-gelE- strains. Could this discrepancy be explained by leaky gelE expression, in the absence of fsrABCD, at levels too low to be detected by qualitative method (i.e., gelatin agar assays)? Assessing protein expression may provide further insights into these differences among the fsrABCD-gelE+ strains and may explain why isolates 32_EF and EF_1008 fail to proliferate.

Line 209: It is not entirely clear whether E. faecalis is replicating within the cytosol, phagosome, or phagolysosome. Could the non-replicating cells be trapped in the phagosome, while the replicating cells have escaped? Including an additional labeling of phagosome-derived vesicle in macrophages with a membrane dye might help provide further insight into this distinction.

Fig. 4C: Based on the localization of GFP-expressing E. faecalis, the non-replicating cells appear punctate, whereas the replicating cells are dispersed along the periphery of the macrophages. It would be helpful if the authors could comment on whether replicating and non-replicating cells are differentially compartmentalized within macrophages.

Line 240: It would be helpful if the authors could examine whether the fsrABCD system is expressed in non-phagocytosed GFP-expressing E. faecalis cells. Interestingly, replicating E. faecalis (GFPhi) comprise only a small proportion (1-5%) compared to the total non-replicating cells (GFPlo) within macrophages. One possibility is that fsrABCD is highly expressed outside of host cells, and that phagocytosis acts as a bottleneck event that dampens expression. This may explain why a significant proportion of E. faecalis cells remain non-replicating and express low levels of fsr and associated effectors.

In addition, it would be informative to know the baseline expression of the fsr system during extracellular replication at 2, 6, and 20 hours (i.e., in the absence of macrophages).

Line 289-296: Similar to the point mentioned earlier, could the large, highly dense intracellular ∆gelE cells reflect an inability to escape from the phagosome, whereas the more dispersed bacterial clusters observed in WT cells represent successful escape?

Line 285, Fig. 5B: The ∆fsrABDC and ∆fsrA mutants appear to egress at levels that surpass the wild-type, despite lacking secreted GelE. Do the authors have qPCR data to assess whether other fsr-regulated factors are differentially expressed in these mutants? This may help clarify whether compensatory virulence factors contribute to the enhanced egress phenotype.

Reviewer #2: No major issues, no additional experiments requested.

Reviewer #3: I don't have major issues with this manuscript.

**Part III – Minor Issues: Editorial and Data Presentation Modifications**

Reviewer #1: Line 74: Please define the fsrABDC locus, briefly describing the roles of fsrA, fsrB, fsrC, and fsrD to orient readers unfamiliar with their functions.

Fig. 1A, B: Please provide an annotation table of genes with their known or predicted functions. Although the genes and associated proteins are described in the introduction, including a table would improve the clarity for the readers. In particular, while the OG1RF 12401::Tn mutant disrupts the gene involved in the phosphotransferase systems PTS8, it is unclear which gene is disrupted in the 12402::Tn mutant.

Line 162: Please clarify in the text that the “fsrABDC–gelE+” strains lack fsrA but retain either an intact or partial fsrBDC.

Fig. 3C: Microscopy images are blurry. Suggest putting in new higher resolution images.

Line 274: At 6 hpi, GelE appears to may a role in host cell lysis. However, by 20 hpi, comparable cytotoxicity is observed in both GelE+ and GelE– strains, as the authors note. Could this later-stage cytotoxicity be attributed to other virulence factors that mediate macrophage lysis and bacterial release (e.g., SprE or additional proteases, cytolysin, Epxs or other pore-forming toxins)?

Fig. 5A: Does complementation restore cytotoxicity to wild-type levels? It would strengthen the conclusions to include the complemented strain (∆gelE::gelE).

Reviewer #2: 1. Clarification about GelE catalytic residues: In Fig. S2, authors state in the legend “Only mutations in homologous residues E346 and E369 of MCP-02 produced stable, non-proteolytic proteases.” Is this by AlphaFold prediction or experimentally tested (and needs a citation)? A clarification in the legend in general could be helpful because it took me a minute to figure out that the chosen mutations for GelE were based on homology to other proteases where the catalytic residues were known.

2. Clarification about the GelE mutants: at line 130-131, it says the E329A mutant is impaired in secretion – is this because there is less protein in the supernatant compared with the supernatants in the other strains in Fig. 1J? Can a loading control be included in Fig. 1J as it was for Fig. S3? Why are the bands different in size and intensity in Fig. 1J (more separation between pro- and processed GelE, more intense) vs Fig. S3 (less separation between pro- and +CTD, and more separation between +CTD and fully processed GelE, less intense bands)?

3. The results section for Fig. 4 needs a little more narrative explanation. For example, it needs to be more explicitly stated that these experiments were performed on a WT strain with GelE expression and therefore has a lower intracellular replication rate than mutants discussed earlier. I was surprised that the number of cells with replicating bacteria were so low until I realized this. I was also confused about why gene expression was not different at 20hrs. Are the authors implying that the bacteria in the GFP-lo cells (which are called non-replicating in the results section, supported by data in Fig. 1) upregulated GelE independent of quorum sensing? Or that at 20 hr p.i. there's no additional replication in the GFP-hi cells that upregulated quorum sensing?

4. In Fig. 5B and 5C, do the authors have an explanation for why the mutants lacking gelE have more (5B) or an equal (5C) number of egressed bacteria at the late time points? Is the high cytotoxicity (and presumably lysis) at the late time points indicated in Fig. 5A an alternative avenue of egress? Address in discussion?

5. Would like a little more elaboration or reorganization in the discussion about selection pressures in the wound site (lines 355-362, 383-400). There were no fitness advantages in the mouse model (Fig. 6), but this is in the absence of antibiotic treatment of the mouse and other human-specific skin microbes that may be present in natural infection, which might be able to apply such selection pressure. Furthermore, are there dis/advantages to dysregulating the expression of other fsr-regulated genes, such as the ones that resulted in lower bacterial recovery shown in Fig. 1? The authors suggest the role of biofilms later on (lines 393-395). Would the combined effect help put pressure on mutating the fsr locus? Or since later in the discussion the authors state that fsr locus and gelE loss are common among isolates from other infection sites, does the mutation occur elsewhere in the body and confer an advantage in a skin wound site? Is it surprising that there’s no clonal expansion of these mutants?

6. Scales on some of the figures (e.g. Fig. 1K, L, Fig. 5A-C) could be adjusted so that difference at 6 hrs p.i. are easier to see.

Reviewer #3: - The authors show that GelE-negative strains exhibit enhanced intracellular replication (Fig. 3A–C). Mechanistically, however, the connection between GelE, the Fsr system, and replication remains unclear. Why do strains lacking these genes replicate more rapidly? Is there a metabolic or biosynthetic burden associated with producing these factors that constrains growth, or does overexpression of these systems actively slow replication? It would be helpful for the authors to elaborate on these possibilities in the Discussion and to propose future experiments that could directly test these ideas.

- Line 238 — I’m confused by the principle of the two-color fluorescence dilution FACS assay. Shouldn’t the eFluor signal decrease through dilution as cells replicate, similar to the assay described in the previous section? Why do the authors state that it remains constant? Please clarify.

- The findings on GelE’s role in mediating E. faecalis egress from infected cells are fascinating, particularly given the implication that GelE may act on one or more host-derived substrates. That said, the impact of infection with the different strains on the host cells feels like the most missing piece of this study. While I don’t think host-focused experiments are necessary at this stage, the Discussion would be strengthened by outlining plausible host substrates that could be targeted by GelE, and by briefly describing how the authors would approach testing these candidate targets in future experiments.

Minor comment:

- Line 105: state what the genes encode for clarity

PLOS authors have the option to publish the peer review history of their article (what does this mean? ). If published, this will include your full peer review and any attached files.

**Do you want your identity to be public for this peer review?** For information about this choice, including consent withdrawal, please see our Privacy Policy .

Reviewer #1: No

Reviewer #2: **Yes:** Cheryl YM Okumura

Reviewer #3: No

**Figure resubmission:**

**Reproducibility:**



---

## [Editor Report · Decision Letter 1]

26 Feb 2026

Dear Professor Kline,

We are pleased to inform you that your manuscript 'Gelatinase regulates the egress of intracellular replicating populations during Enterococcus faecalis infection' has been provisionally accepted for publication in PLOS Pathogens.

Best regards,

Breck A. Duerkop

Academic Editor

PLOS Pathogens

Alice Prince

Section Editor

PLOS Pathogens

Sumita Bhaduri-McIntosh

Editor-in-Chief

PLOS Pathogens

orcid.org/0000-0003-2946-9497

Michael Malim

Editor-in-Chief

PLOS Pathogens

orcid.org/0000-0002-7699-2064

Thank you for submitting your revised manuscript. I appreciate your comprehensive responses to the reviewer's previous critiques and suggestions. It is my assessment that you have sufficiently addressed them and that this manuscript can be accepted in its current form for publication. Congratulations!
---

## [Editor Report · Acceptance letter]

Dear Professor Kline,

We are delighted to inform you that your manuscript, "Gelatinase regulates the egress of intracellular replicating populations during Enterococcus faecalis infection," has been formally accepted for publication in PLOS Pathogens.

Best regards,

Sumita Bhaduri-McIntosh

Editor-in-Chief

PLOS Pathogens

orcid.org/0000-0003-2946-9497

Michael Malim

Editor-in-Chief

PLOS Pathogens

orcid.org/0000-0002-7699-2064